# Offline Equilibrium Finding in Extensive-form Games: Datasets, Methods, and Analysis

## Abstract

Recently, offline reinforcement learning (Offline RL) has emerged as a promising paradigm for solving real-world decision-making problems using pre-collected datasets. However, its application in game theory remains largely unexplored. To bridge this gap, we introduce *offline equilibrium finding* (Offline EF) in extensive-form games (EFGs), which aims to compute equilibrium strategies from offline datasets. Offline EF faces three key challenges: the lack of benchmark datasets, the difficulty of deriving equilibrium strategies without access to all action profiles, and the impact of dataset quality on effectiveness. To tackle these challenges, we first construct diverse offline datasets covering a wide range of games to support algorithm evaluation. Then, we propose BOMB, a novel framework that integrates behavior cloning within a model-based method, enabling seamless adaptation of online equilibrium-finding algorithms to the offline setting. Furthermore, we provide a comprehensive theoretical analysis of BOMB, offering performance guarantees across various offline datasets. Extensive experimental results show that BOMB not only outperforms traditional offline RL methods but also achieves highly efficient equilibrium computation in offline settings.

## 1 Introduction

Extensive-form games (EFGs) offer a powerful framework for modeling interactions among multiple players in stochastic and imperfect information settings (Nisan et al., 2007). The canonical solution concept is the Nash Equilibrium (NE), where no player can unilaterally deviate to increase their own utility. There are various methods designed for solving EFGs, including linear programming (Shoham & Leyton-Brown, 2008), double-oracle algorithms (McMahan et al., 2003), counterfactual regret minimization (CFR) (Zinkevich et al., 2007), and policy-space response oracles (PSRO) (Lanctot et al., 2017). These techniques have been successfully applied to real-world large-scale EFGs, such as pursuit-evasion games (Xue et al., 2021; Li et al., 2023), poker games (Brown & Sandholm, 2018; 2019; Zha et al., 2021), and Stratego (Perolat et al., 2022). Despite their successes, existing algorithms for equilibrium finding rely on interactions with the game environment or access to an accurate simulator. For example, CFR-based algorithms require traversing the game tree to compute regret values, while PSRO and its variants depend on simulations to compute the best response oracle and estimate entries in the meta-game. We refer to this paradigm as "*online equilibrium finding*". However, in many real-world applications, including sports games (Liu et al., 2022), network intrusion detection (Khraisat et al., 2019), and automated negotiations (Kiruthika et al., 2020), interacting with the environment in real time is often prohibitively costly or inefficient, while constructing an accurate simulator is typically infeasible. In such cases, offline learning offers a practical and sample-efficient alternative for equilibrium finding.

Recent efforts have been made to formalize the offline learning paradigm within game settings. For example, StarCraft II Unplugged (Mathieu et al., 2021) offers a dataset of human gameplays for a two-player zero-sum symmetric game. Some works (Cui & Du, 2022; Zhong et al., 2022) investigate the properties of offline datasets necessary for successfully inferring NEs in two-player zero-sum Markov games. However, existing work on offline learning in games has focused primarily on Markov games, leaving extensive-form games largely unexplored. Moreover, to the best of our knowledge, there has been no dedicated study addressing the offline setting in multi-player games. More fundamentally, the offline learning paradigm itself remains underdefined and insufficiently studied within the broader context of game-theoretic learning.

Figure 1: Comparison between online and offline equilibrium finding

To bridge this gap, we propose the novel ***offline equilibrium finding*** (Offline EF) paradigm, which computes equilibrium strategies using offline datasets. Offline EF faces several key challenges. First, the lack of comprehensive benchmarking standards complicates the evaluation and comparison of algorithms, making it difficult to objectively measure progress in the field. Second, accurately computing or approximating equilibrium strategies solely from offline datasets is inherently challenging. Specifically, data from only a subset of action profiles often cannot determine proximity to an equilibrium strategy, as identifying equilibria requires referencing all other potential action profiles (Cui & Du, 2022). Third, the quality and completeness of offline datasets significantly affect the effectiveness of derived strategies. Offline datasets rarely cover all possible game states, and this limited coverage hampers the algorithm's ability to generalize effectively from the available data.

This work provides a comprehensive investigation of offline EF with four key contributions: i) We curate a diverse collection of offline datasets, including random, expert, learning, and hybrid datasets across various extensive-form games; ii) we propose the BOMB framework, which combines behavior cloning with a model-based method, featuring a novel parameter estimation approach and the ability to incorporate any online EF algorithm (e.g., CFR) into the offline context; iii) we offer a thorough theoretical analysis for BOMB, providing performance guarantees across different datasets; and iv) we validate the effectiveness of BOMB through extensive experiments, demonstrating its capability to compute equilibrium strategies from offline datasets.

## 2 PRELIMINARIES

**Imperfect-Information Extensive-Form Games(IIEFGs).** An IIEFG can be represented as a tuple, formally, $\mathcal{G} = (N, H, A, P, \mathcal{I}, u)$ (Shoham & Leyton-Brown, 2008). $N = \{1, ..., n\}$ is the set of players, and $H$ is the set of histories, representing all possible action sequences, including the root node of the game tree (empty sequence $\emptyset$) and all prefixes of sequences in $H$. The set of terminal histories, represented by $Z$, represents the set of game outcomes, $Z \subseteq H$. $A(h) = \{a : (h, a) \in H\}$ is the set of available actions at any non-terminal history $h \in H \setminus Z$. $P$ is the player function mapping non-terminal histories to players, i.e., $P(h) \mapsto N \cup \{c\}, \forall h \in H \setminus Z$, where $c$ represents the "chance player" for stochastic events outside players' control. $\mathcal{I}$ is the set of information sets, forming a partition over histories where player $i$ takes actions, such that player $i$ cannot distinguish these histories within the same information set $I_i$. Each information set $I_i \in \mathcal{I}_i$ represents one decision point which means that $P(h_1) = P(h_2)$ and $A(h_1) = A(h_2)$ for any $h_1, h_2 \in I_i$. For convenience, $A(I_i)$ and $P(I_i)$ denote the action set $A(h)$ and player $P(h)$ for any $h \in I_i$. For each player $i$, a utility function $u_i : Z \to \mathbb{R}$ defines the payoff for every terminal history. The behavior strategy of player $i$, $\sigma_i$, maps each information set of $i$ to a probability distribution over $A(I_i)$, with $\Sigma_i$ denoting the set of all strategies for player $i$. A strategy $\sigma_i$ is *pure* if it always selects a single action (i.e., $\sigma_i(I_i, a) \in \{0, 1\}$), *mixed* if it assigns probabilities in $[0, 1]$, and *fully mixed* if all probabilities are strictly positive. A strategy profile $\sigma = (\sigma_1, \sigma_2, ..., \sigma_n)$ is a tuple of strategies for all players, with $\sigma_{-i}$ referring to all strategies except $\sigma_i$. The reaching probability of a history $h$ under $\sigma$ is $\pi^\sigma(h) = \prod_{i \in N \cup \{c\}} \pi_i^\sigma(h)$. Given $\sigma$, the expected payoff for player $i$ is $u_i(\sigma) = \sum_{z \in Z} \pi^\sigma(z) u_i(z)$, summing over the terminal nodes.

**Solution Concepts.** The common solution concept for IIEFGs is Nash equilibrium (NE) (Nash, 1950), where no player can increase their utility by unilaterally deviating. Formally, a strategy profile $\sigma^*$ forms an NE if it satisfies $\forall i \in N, u_i(\sigma^*) = \max_{\sigma_i' \in \Sigma_i} u_i(\sigma_i', \sigma_{-i}^*)$. To measure the distance from NE in $N$-player games, we use NASHCONV($\sigma$) as the metric, where NASHCONV($\sigma$) =

$\sum_{i \in N} \text{NASHCONV}_i(\sigma) = \sum_{i \in N}(\max_{\sigma_i'} u_i(\sigma_i', \sigma_{-i}) - u_i(\sigma))$. When $\text{NASHCONV}(\sigma) = 0$, then $\sigma$ is an NE. In $N$-player general-sum games, (Coarse) Correlated Equilibrium ((C)CE) is also a common solution concept (Aumann, 1987). A CE is a joint mixed strategy in which no player has an incentive to deviate. Let $S_i$ be the strategy space for player $i$. A strategy profile $\sigma^*$ forms a CCE if $\forall i \in N, s_i \in S_i, u_i(\sigma^*) \ge u_i(s_i, \sigma_{-i}^*)$ where $\sigma_{-i}^*$ is the marginal distribution of $\sigma^*$ on strategy space $S_{-i}$. The (C)CE Gap Sum is adopted to measure the distance from (C)CE (Marris et al., 2021).

**Why Existing Methods Fail?** Offline reinforcement learning (RL) primarily targets single-agent settings (Levine et al., 2020) and is inherently incapable of computing equilibrium strategies in games. Opponent modeling (OM) focuses on predicting the behavioral strategies of opponents (He et al., 2016), but it is also designed to compute the best response strategy for one player rather than equilibrium strategies. Furthermore, OM requires access to the game environment, making it unsuitable for offline EF.

| Methods | Work w/o env | Converge to equilibrium |
|---|---|---|
| Offline RL | ✓ | ✗ |
| OM | ✗ | ✗ |
| Online EF | ✗ | ✓ |

Table 1: Issues of Existing Methods.

Widely used equilibrium-finding algorithms, such as no-regret methods (e.g., CFR (Zinkevich et al., 2007)) and empirical game-theoretic analysis methods(e.g., PSRO (Lanctot et al., 2017)), fundamentally rely on repeated interactions with the game environments or access to an accurate simulator. These methods, referred to as "Online EF", are therefore not directly applicable to the offline setting. A comparison of existing methods is provided in Tab. 1, with further discussion in App. B.

**Problem Statement.** To facilitate the widespread application of game theory, we extend the offline learning framework to extensive-form games and introduce the *offline equilibrium finding* paradigm.

**Definition 2.1** (Offline EF). Let $\mathcal{D}$ be an offline dataset of an IIEFG $\mathcal{G}$, generated by an unknown behavior strategy $\sigma$. The objective of the *offline equilibrium finding* paradigm is to derive a strategy profile $\widehat{\sigma}$ from $\mathcal{D}$ that minimizes the gap from the equilibrium strategy $\sigma^*$. Formally, $\widehat{\sigma} = \arg\min_{\sigma' \in \Sigma} \text{GAP}(\sigma', \sigma^*)$, where $\text{GAP}(\cdot)$ is a metric function quantifying the gap between $\sigma'$ and the equilibrium strategy. If $\text{GAP}(\sigma', \sigma^*) \le \epsilon$, then $\sigma'$ is said to be an $\epsilon$-equilibrium strategy.

Building on the definition of the offline EF paradigm, we can instantiate it by defining a metric for the gap from the equilibrium strategy, such as the $\text{NASHCONV}$ for NE (Nash, 1950) and (C)CE Gap Sum for (C)CE (Aumann, 1987). While offline EF shares some similarities with offline RL, it introduces unique challenges. First, unlike offline RL, which aims to compute an optimal strategy (Levine et al., 2020), offline EF seeks to achieve an equilibrium strategy. This requires iterative processes for best response calculations, adding significant complexities. Second, offline EF inherently involves at least two players, making the game dynamics highly sensitive to distribution shifts and uncertainties – a stark contrast to offline RL. Third, in offline RL, the data from two actions may suffice to determine which action is better. However, in offline EF, simply comparing data from two action tuples is insufficient to identify which action tuple is closer to an equilibrium strategy, as equilibrium identification requires additional action tuples for references (Cui & Du, 2022).

## 3 DATASETS

Datasets play a pivotal role in offline learning, however, there are no publicly available datasets specifically tailored for the offline EF paradigm. Consequently, we outline our methods to collect datasets at different expert levels that will serve as a basis for advancing offline EF research.

**Formats.** Before discussing data collection methods, it is essential to first define the data formats of an offline EF dataset for IIEFGs. The dataset is represented as $\mathcal{D} = (s_t, a_t, s_{t+1}, u_{t+1}, d_{t+1})$. $s_t$ and $s_{t+1}$ represent the game states at time step $t$ and $t+1$ from a game-level perspective. We slightly overload the term "state" here, using it in a sense distinct from the previously defined notions of history and information set. Specifically, $s_t$ includes all information at $t$, including the information sets for each player $(I_1^t, I_2^t, ..., I_n^t)$, additional game-level information outside the players' control $(GI)$, the acting player at $t$ $(p^t)$, and the available action set for the acting player $(A(I_{p^t}^t))$. Formally, $s_t = (I_1^t, I_2^t, ..., I_n^t, GI, p^t, A(I_{p^t}^t))$, where $p^t$ may also be a chance player $c$. $a_t$ is the action taken at $t$. $u_{t+1} = (u_1^{t+1}, u_2^{t+1}, ..., u_n^{t+1})$ represents the utilities for all players at time step $t+1$. Finally, $d_{t+1}$ indicates whether the game ends at state $s_{t+1}$, with a value of 1 if the game ends and 0 otherwise.

**Collecting Methods.** Similar to offline RL, offline EF datasets must be diverse to support robust development and evaluation. Prior offline RL benchmarks (Fujimoto et al., 2019; Gulcehre et al., 2020) often use data from online RL runs, while D4RL (Fu et al., 2020) incorporates human demonstrations, exploratory agents, and hand-coded controllers. Inspired by these, we propose three data collection methods for offline EF at varying expertise levels. The first one, referred to as the *random method*, involves players adopting uniform strategies, generating data that mimics novice exploratory behavior. The second method, the *learning-based method*, collects intermediate game interactions using equilibrium finding algorithms, such as CFR (Zinkevich et al., 2007) or PSRO (Lanctot et al., 2017), reflecting the process of skill improvement. The final method, the *expert method*, generates data by having players follow equilibrium strategies, capturing expert-level gameplay. Additionally, a hybrid approach combines random and expert datasets in varying proportions, enhancing realism and diversity to create more comprehensive benchmarks.

**Statistics of Datasets.** We developed a benchmark dataset for offline EF using these outlined collection methods across **eight** commonly used IIEFGs, as depicted in Fig. 2. The dataset contains approximately **3.8 million** data points, occupying about **11GB** of memory. For each game, we generated three distinct types of datasets: Expert, random, and learning. The proportions of each dataset are visually detailed and comprehensive statistics on the distribution of these datasets are detailed further in App. C.2.

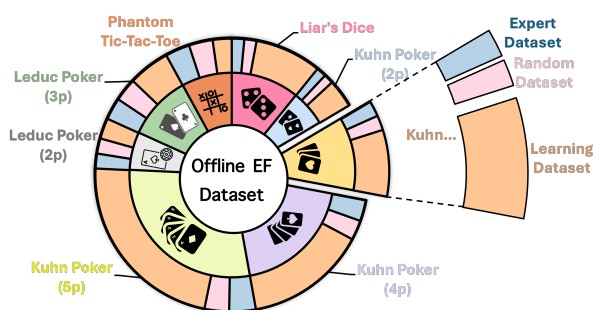

Figure 2: Dataset of Offline EF.

## 4 BOMB

Inspiring by offline RL algorithms, there are two main directions to develop the algorithmic framework for offline EF: i) behavior cloning (BC) (Fujimoto & Gu, 2021), which imitates the strategies used to generate the offline data with additional exploration, and ii) model-based methods (Yu et al., 2020; Kidambi et al., 2020), which learns a world model from the offline dataset and then derive strategies from the world model. However, BC may fail in offline EF when the dataset is generated by random policies, as such strategies are often far from equilibrium. Similarly, model-based methods struggle in offline EF when the dataset originates from equilibrium strategies, which often cover only a limited portion of the game states. To mitigate these issues, we propose BOMB, which combines **B**ehavior cl**O**ning and **M**odel-**B**ased method for the offline EF paradigm.

### 4.1 BOMB FRAMEWORK

**BOMB.** Alg. 1 outlines the BOMB framework. Given an offline dataset $\mathcal{D}$, BOMB starts by training a policy $\sigma_\theta$ using behavior cloning (BC) (Line 2). The dataset $\mathcal{D} = (s_t, a_t, s_{t+1}, u_{t+1}, d_{t+1})$ contains game sate $s_t = (I_1^t, I_2^t, ..., I_n^t, GI, p^t, A(I_{p^t}^t))$. Since $\sigma_\theta$ mimics the behavior strategy, only $I_{p^t}^t$ and the corresponding action $a_t$ are used for training. The training

---

**Algorithm 1** BOMB Framework

1: **Input:** an offline dataset $\mathcal{D}$
2: Train policy $\sigma_\theta$ based on $\mathcal{D}$ using BC technique;
3: Train an environment model $E_{\theta_e}$ based on $\mathcal{D}$;
4: Learn $\sigma_{mb}$ policy using any EF algorithm on $E_{\theta_e}$;
5: Select $\alpha$ using parameter estimation method;
6: $\sigma = \alpha \cdot \sigma_\theta + (1 - \alpha) \cdot \sigma_{mb}$;
7: **Output:** Policy $\sigma$

---

loss is the cross-entropy loss: $\mathcal{L}_{bc} = -\mathbb{E}_{(I_{p^t}^t, a_t) \sim \mathcal{D}}[a_t \cdot \log(\sigma(I_{p^t}^t; \theta))]$. Additionally, inspired by model-based offline RL algorithms (Kidambi et al., 2020; Yu et al., 2020), a dynamic environment model $E_{\theta_e}$ is trained based on dataset $\mathcal{D}$ (Lines 3). The model simulates the real environment and is used to derive the model-based (MB) policy $\sigma_{mb}$ using any online EF algorithm, such as PSRO (Lanctot et al., 2017) and CFR (Zinkevich et al., 2007) (Line 4). Specifically, we use the game state $s_t$ and action $a_t$ as inputs, with the subsequent game state $s_{t+1}$, reward $u_{t+1}$, and termination variable $d_{t+1}$ serving as labels. The model is trained using stochastic gradient descent

(SGD) with the mean squared error loss: $\mathcal{L}_{env} = \mathbb{E}_{\mathcal{D}}[\mathbf{MSE}((s_{t+1}, u_{t+1}, d_{t+1}), E(s_t, a_t; \theta_e))]$. The final policy combines the BC policy $\sigma_\theta$ and MB policy $\sigma_{mb}$ using a weighted combination: $\sigma = \alpha\sigma_\theta + (1 - \alpha)\sigma_{mb}$, where $\alpha$ is the weight of the BC policy (Lines 5-6). The method for estimating $\alpha$ is described in the following.

**Estimation of Parameter $\alpha$.** We propose three methods of estimating the parameter $\alpha$. The simplest is the random selection method, where a value is randomly chosen from the interval $[0, 1]$. This method is fully offline and easy to implement, but lacks guarantees for finding the most effective combined strategy. The second method is the grid search method, where we define a set of 11 candidate values for $\alpha$, i.e., $\{0, 0.1, ..., 1\}$. These values are used to configure combined policies, which are then tested in a real environment. The value of $\alpha$ that results in the smallest gap from the equilibrium strategy is selected as optimal. This method achieves the best performance by systematically exploring possible values and is commonly used in offline RL for fine-tuning param-

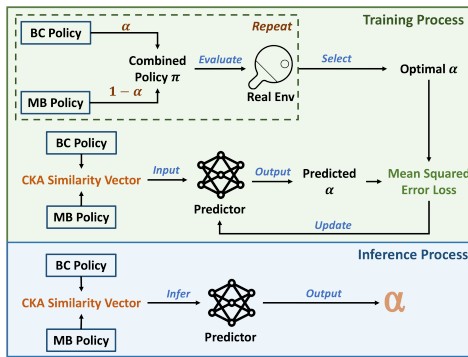

Figure 3: Learning-based estimation method.

eters through online interactions (Kalashnikov et al., 2018; Lee et al., 2022). However, it requires online evaluations, making it less practical in fully offline settings. To enable fully offline while retaining good performance, we propose a learning-based method, as depicted in Fig. 3. This method involves training a predictor to estimate $\alpha$ based on the relationship between the BC and the MB policies. We first use the grid search method to get optimal $\alpha$ values as labels. The predictor takes a similarity vector derived from the centered kernel alignment (CKA) (Kornblith et al., 2019) between the BC and the MB policies and outputs the estimated $\alpha$. The predictor is trained in one game where online interactions are feasible and can then be reused in similar games. Although the predictor provides only an approximate optimal parameter value, it eliminates the need for further online interactions once trained, offering a practical and efficient solution for fully offline settings.

**Advantages of BOMB.** First, by combining BC and MB methods, BOMB can work on the datasets collected by any strategies, whether random or equilibrium strategies. This versatility ensures that BOMB can adapt to diverse offline data sources. Second, with the learned environment model for games, BOMB seamlessly integrates online EF algorithms, enabling it to generalize across different equilibrium types. For example, we adapt PSRO (Lanctot et al., 2017) and Deep CFR (Brown et al., 2019) methods to compute NE, referred to as MB-PSRO and MB-CFR, respectively. Additionally, we adapt JPSRO method (Marris et al., 2021) (MB-JPSRO) for computing (C)CE. This flexibility allows BOMB to address a wide range of equilibrium-finding tasks. Third, BOMB is game-agnostic, meaning it can learn game rules directly from offline datasets without requiring prior knowledge of the game itself. This characteristic is similar to the advantage of MuZero (Schrittwieser et al., 2020). This makes BOMB applicable to a broad array of games.

## 4.2 THEORETICAL ANALYSIS

In the offline RL area, dataset coverage over the optimal policy is typically sufficient for offline learning (Rashidinejad et al., 2021; Xie et al., 2021). However, in the offline EF paradigm, the assumption that the equilibrium strategy generates the dataset is not sufficient for computing equilibrium strategies offline. This insufficiency is illustrated by the counter-example in Fig. 4. In this example, we can easily get NE strategy, $\sigma^* = (\sigma_1^*, \sigma_2^*) = (\{I_1 : a_1\}, \{I_2 : b_2\})$. If this equilibrium strategy is used to generate the offline dataset $\mathcal{D}$, the dataset $\mathcal{D}$ would only contain the single data point $((I_1^{t_1} = I_1, I_2^{t_1} = \emptyset, \emptyset, 1, \{a_1, a_2\}), a_1, (I_1^{t_2} = I_1a_1, I_2^{t_2} = \emptyset, \emptyset, -1, \emptyset), (0, 0), 1)$. Clearly, the dataset $\mathcal{D}$ lacks any information about Player 2, making it insufficient for computing the NE strategy. Similarly, the assumption that the offline dataset covers the equilibrium strategy is also insufficient for the offline EF paradigm, as we prove in App. D.1. In

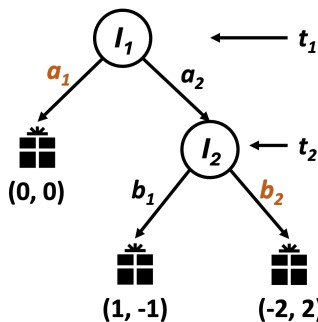

Figure 4: Game example.

this section, we outline the necessary and sufficient conditions for dataset coverage to guarantee the convergence of our methods in IIEFGs with perfect recall. To achieve this, we introduce two key concepts of dataset coverage: uniform coverage and equilibrium coverage.

**Definition 4.1.** An offline dataset $\mathcal{D}$ is said to be a **uniform coverage** of an IIEFG $\mathcal{G}$ if and only if it covers all possible state-action pairs in the game. Formally, it covers $(s_t, a_t, s_{t+1}, u_{t+1}, d_{t+1}), \forall s_t, a_t \in A(s_t)$ and $s_{t+1} \in T(s_t, a_t)$ where $T$ is the transition function.

**Definition 4.2.** An offline dataset $\mathcal{D}$ is said to be an $\epsilon$-**equilibrium coverage** of an IIEFG $\mathcal{G}$ if and only if its underlying behavior strategy $\sigma_{\mathcal{D}}$ satisfies: $\text{GAP}(\sigma_{\mathcal{D}}, \sigma^*) < \epsilon$, where $\sigma_{\mathcal{D}}$ is defined as: $\sigma_{\mathcal{D}}(s_t, a_t) = \frac{C(s_t, a_t)}{C(s_t)}$ and $\sigma_D(s_t, a_t) > 0$ for all $s_t$ and $a_t \in A(s_t)$, with $C(s_t, a_t)$ and $C(s_t)$ denoting the counts of occurrences of state-acion pair $(s_t, a_t)$ and the state $s_t$ in $\mathcal{D}$, respectively.

Building on these two dataset coverage definitions, we now discuss the conditions required for our method to achieve convergence. To facilitate this analysis, we introduce an assumption on the approximation error incurred when training neural networks.

**Assumption 4.3.** The training error of neural networks in BOMB is assumed to be bounded by an arbitrarily small $\epsilon$ given a sufficiently large and diverse dataset.

To support this assumption, we provided a general generalization bound for the training error under a dataset with size $m$ in App. D.2. Then we present our result as follows.

**Theorem 4.4.** *Let $\sigma_{MB(\mathcal{D})}$ be the strategy learned by **MB algorithm** using the offline dataset $\mathcal{D}$ with sufficient data under Assump. 4.3. Then, $\sigma_{MB(\mathcal{D})}$ is guaranteed to be an $\epsilon$-equilibrium strategy of IIEFG $\mathcal{G}$ if $\mathcal{D}$ is a uniform coverage of $\mathcal{G}$ and $\sigma_{MB(\mathcal{D})}$ is an $\epsilon$-equilibrium strategy of the trained environment used in the MB algorithm. If either condition fails, the guarantee may no longer hold.*

*Sketch Proof.* Under Assumption 4.3, the error incurred during training of the environment game model from $\mathcal{D}$ is negligible. As a result, the trained environment game model is identical to the original game $\mathcal{G}$, provided that the dataset $\mathcal{D}$ offers full coverage of all state transitions. Consequently, if $\sigma_{MB(\mathcal{D})}$ is an $\epsilon$-equilibrium strategy for the trained environment game model, it is also an $\epsilon$-equilibrium strategy for the original game $\mathcal{G}$. Any slight deviation from these conditions would invalidate the convergence guarantee. A complete proof of this result is detailed in App. D.1. $\square$

**Theorem 4.5.** *Let $\sigma_{BC(\mathcal{D})}$ be the strategy learned by **BC algorithm** using the offline dataset $\mathcal{D}$ with sufficient data under Assump. 4.3. Then, $\sigma_{BC(\mathcal{D})}$ is guaranteed to be an $\epsilon$-equilibrium strategy of IIEFG $\mathcal{G}$ if $\mathcal{D}$ is an $\epsilon$-equilibrium coverage of $\mathcal{G}$. Otherwise, the guarantee may no longer hold.*

*Sketch Proof.* According to Assumption 4.3, the error in training the behavior cloning strategy $\sigma_{BC(\mathcal{D})}$ from the dataset $\mathcal{D}$ is negligible. As a result, the behavior cloning process ensures that $\sigma_{BC(\mathcal{D})}$ is identical to the behavior strategy underlying $\mathcal{D}$, i.e., $\sigma_{BC(\mathcal{D})} = \sigma_{\mathcal{D}}$. Consequently, if $\mathcal{D}$ provides $\epsilon$-equilibrium coverage of $\mathcal{G}$, then $\sigma_{BC(\mathcal{D})}$ is guaranteed to be an $\epsilon$-equilibrium strategy for the IIEFG $\mathcal{G}$, since $\text{GAP}(\sigma_{\mathcal{D}}, \sigma^*) < \epsilon$ implies $\text{GAP}(\sigma_{BC(\mathcal{D})}, \sigma^*) < \epsilon$. Any deviation from these conditions would invalidate the convergence guarantee. The full proof is provided in App. D.1. $\square$

Building on the insights provided by the preceding two theorems, we propose the following theorem to characterize the performance of BOMB in a general case where the offline dataset is generated by an unknown strategy profile. The full proof and detailed derivation are provided in App. D.1.

**Theorem 4.6.** *Let $\sigma_{BOMB(\mathcal{D})}$ represent the strategy profile learned by **BOMB algorithm** based on the offline dataset $\mathcal{D}$ with sufficient data under Assumption 4.3, $\sigma_{\mathcal{D}}$ represent the underlying behavior strategy of $\mathcal{D}$ and $\sigma^*$ represent the equilibrium strategy of IIEFG $\mathcal{G}$. Then the gap between $\sigma_{BOMB(\mathcal{D})}$ and $\sigma^*$ is guaranteed to be at most equal to, or smaller than, the gap between $\sigma_{\mathcal{D}}$ and $\sigma^*$. Formally, $\text{GAP}(\sigma_{BOMB(\mathcal{D})}, \sigma^*) \leq \text{GAP}(\sigma_{\mathcal{D}}, \sigma^*)$.*

To analyze the performance of our algorithm in real-world scenarios, we first examine the offline datasets generated for the offline EF paradigm. Based on dataset collection procedures, we observe that the random dataset can be considered as a uniform coverage of the game $\mathcal{G}$ when the dataset size is sufficiently large. This is because the random dataset is collected using a uniform strategy, ensuring that every action is adequately sampled as the dataset grows. On the other hand, the expert dataset can be considered as an $\epsilon$-equilibrium coverage of IIEFG $\mathcal{G}$, where $\epsilon$ decreases as the dataset

size increases. Since the expert dataset is generated by an equilibrium strategy, a larger dataset size allows the underlying behavior strategy of the dataset to approximate the equilibrium strategy more closely, thereby reducing $\epsilon$. These dataset properties ensure that the theoretical guarantees of our algorithm hold for both random and expert datasets, as shown in the following experimental results.

## 5    EXPERIMENTAL RESULTS

To evaluate the performance of BOMB, we conduct the following experiments: i) we compare BOMB with two offline RL algorithms; ii) we evaluate the performance of different estimation methods for parameter $\alpha$; and iii) we run BOMB on various offline datasets to assess its capability in computing different equilibrium strategies, including NE and CCE, across different game scenarios.

We use OpenSpiel[1] (Lanctot et al., 2019) as our experimental platform, which provides a well-established collection of environments and algorithms for game research, thereby facilitating future replicability. For our experiments, we select several widely used games, including poker games, Liar's Dice, and Phantom Tic-Tac-Toe (Lisý et al., 2015; Brown et al., 2019). The experiments are conducted on a workstation equipped with a ten-core 3.3GHz Intel i9-9820X CPU and NVIDIA RTX 2080Ti GPU. All results are averaged over three seeds, and error bars are also reported.

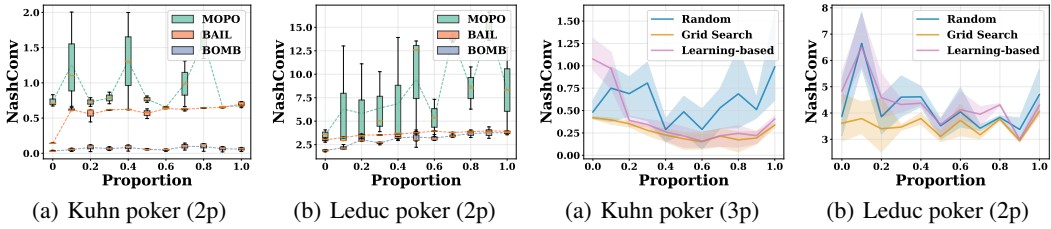

| (a) Kuhn poker (2p) | (b) Leduc poker (2p) | (a) Kuhn poker (3p) | (b) Leduc poker (2p) |

Figure 5: Comparison with offline RL.    Figure 6: Results of different estimation methods.

**RQ1:** *Can BOMB outperform offline RL methods?*

To support this claim that offline RL algorithms are insufficient for the offline EF paradigm, we select two representative algorithms: the model-free algorithm–Best-Action Imitation Learning (BAIL) (Chen et al., 2020) and the model-based algorithm–Model-based Offline Policy Optimization (MOPO) (Yu et al., 2020). The comparison results, shown in Fig. 5, are conducted on two-player Kuhn poker and Leduc poker games under hybrid datasets. The x-axis represents the proportion of data from the random datasets within the hybrid dataset. When the proportion is zero, the hybrid dataset is equivalent to the expert dataset; when the ratio is one, it reduces to the random dataset. The results show that BOMB consistently outperforms both offline RL algorithms across all cases, regardless of the dataset composition. This highlights the limitations of traditional offline RL algorithms in addressing the offline EF paradigm and underscores the effectiveness of BOMB.

**RQ2:** *How do different methods for estimating the parameter $\alpha$ perform?*

We evaluate the three proposed parameter estimation methods through experiments on poker games. For the learning-based method, the parameter predictor was trained on the two-player Kuhn poker game and tested on other poker games. The performance results, shown in Fig. 6, include three-player Kuhn poker and two-player Leduc poker games. These results show that the grid search method consistently achieves the best performance across games, and the learning-based method performs comparably to the grid search method on the three-player Kuhn poker but slightly worse on the two-player Leduc poker game, suggesting that its performance depends on the similarity between the test game and the game used for training. Interestingly, the random method performs surprisingly well in many cases, indicating that even simple combinations can work effectively. For the remaining experiments, we use the grid search method as the parameter estimation method.

**RQ3:** *Can the BOMB framework compute NE?*

To answer this question, we conduct extensive experiments covering two-player games, multi-player games, and real-world scenarios simulated using learning datasets. This comprehensive evaluation ensures a thorough assessment of our method's performance in computing the NE strategy.

---

[1]https://github.com/deepmind/open_spiel

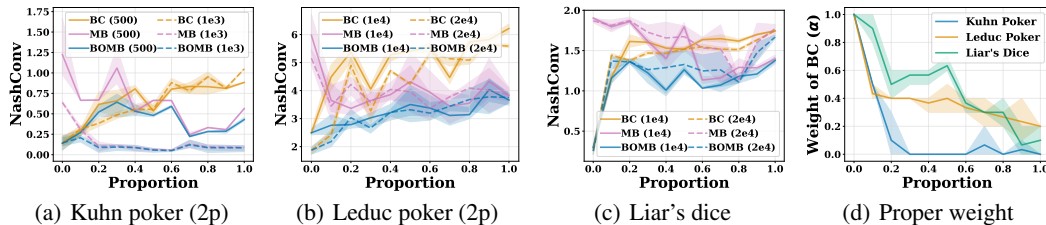

Figure 7: Experimental results on computing NE in two-player games.

**Two-Player Cases.** We first move to evaluate the performance of our algorithm, BOMB, in computing the NE strategy. In addition to performing BOMB, we also assess the individual performance of its components: the behavior cloning (BC) technique and the model-based (MB) algorithm. This evaluation highlights the strengths and weaknesses of each component while offering a comprehensive understanding of the efficacy of BOMB in computing equilibrium strategies offline. Figs. 7(a)-7(c) present results on two-player games under offline datasets with different sizes. For NE computation, we use either MB-CFR or MB-PSRO, and the MB framework's performance is shown to be independent of the specific algorithm used (details in App. F). As the proportion of the random dataset increases, the performance of BC decreases, whereas MB shows a slight improvement. Additionally, we observe that as the size of offline data increases, the improvement of the BC's performance is not significant and the MB's performance improves. It means that the performance of BC mainly depends on the quality of the behavior policy generating the dataset, and the performance of MB relies on the accuracy of the environment model relative to the actual environment. Across all cases, BOMB outperforms both BC and MB methods, demonstrating its ability to compute NE strategies for two-player games. This underscores the effectiveness of combining BC and MB methods in the offline EF paradigm. To further analyze the performance of the BOMB method in two-player games, we plot the parameter $\alpha$ for these combined policies, as shown in Fig. 7(d). The results reveal that as the proportion of the random dataset increases, the weight of the BC policy decreases. This indicates that the BC policy performs better with expert datasets, while the MB policy excels with random datasets. These findings highlight the complementary strengths of BC and MB, further validating the effectiveness of the BOMB framework.

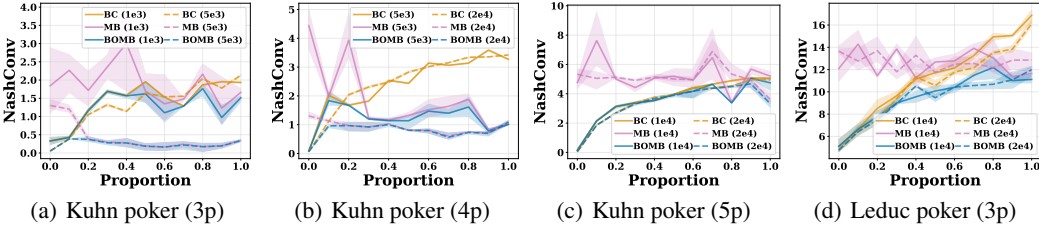

Figure 8: Experimental results on computing NE in multi-player games.

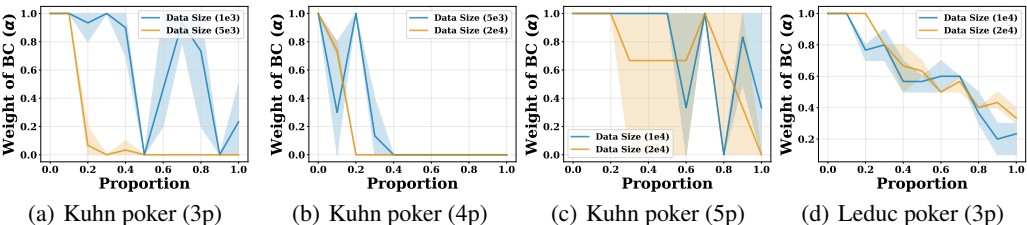

Figure 9: Proper weight of BC policy in multi-player games.

**Multi-Player Cases.** We conduct experiments on multi-player games to evaluate the performance of BOMB in computing NE strategies across several multi-player games, as illustrated in Fig. 8 and App. F. These results show that BOMB consistently performs as well as or better than both BC and MB algorithms, similar to the findings in two-player cases. However, as the proportion of the random dataset increases, the performance of BC decreases, while MB exhibits instability with a slight downward trend. Notably, for multi-player games, it is PPAD-hard to compute NEs. Thus, CFR-

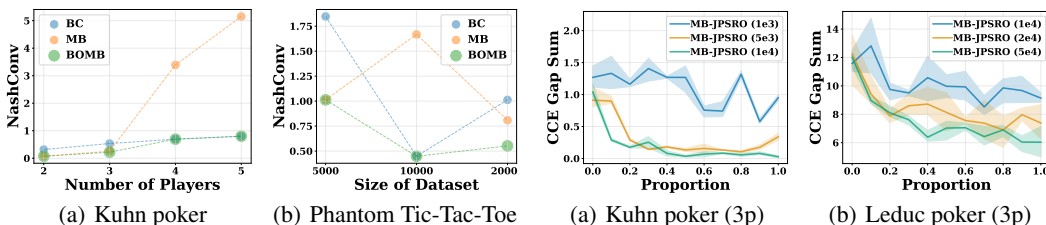

Figure 10: Results on learning dataset.            Figure 11: Results on computing CCE.

based algorithms do not guarantee convergence to NE in multi-player games, and the performance of MB may be influenced. Consequently, the underperformance of the MB algorithm may stem from either an inadequately trained environment game model or inherent limitations of NE computation in multi-player settings. These observations highlight critical challenges for offline EF in multi-player games, particularly the need to develop more effective equilibrium-finding algorithms and train an accurate environment game model. Addressing these challenges is essential for advancing offline EF in multi-player games. The appropriate weights of the BC policy ($\alpha$) within BOMB across different hybrid datasets are shown in Fig. 9 and App. F. In the three-player Kuhn poker game, the weight of the BC policy quickly drops to zero as the proportion of the random dataset increases, indicating that the MB method generally outperforms the BC method except when the random dataset proportion is low. Conversely, in the three-player Leduc poker game, the BC policy retains a high weight in most cases, except when the proportion of the random dataset is high. This suggests the MB method struggles in these games due to challenges in learning an approximate equilibrium strategy. The difficulty stems from the inherent complexity of multi-player games, where the development of effective equilibrium-finding algorithms and the training of accurate game models are challenging.

**Simulating Real-World Cases.** We also conduct experiments on the learning dataset, which closely approximates real-world conditions. Fig. 10(a) shows the results of Kuhn poker games with different numbers of players and Fig. 10(b) shows the results of Phantom Tic-Tac-Toe under datasets with different sizes. It indicates that given an offline dataset generated by an unknown strategy, BOMB can also perform better than BC and MB in approximating the NE strategy.

**RQ4:** *Can the BOMB framework compute CCE?*

We evaluate the performance of the model-based method in computing the CCE strategy. The BC technique and the BOMB framework are not used for CCE computation since the offline dataset is collected using independent strategies for each player rather than a joint strategy. The results of applying the MB-JPSRO algorithm on three-player Kuhn poker and Leduc poker games are presented in Fig. 11. The results show that as the size of the offline data increases, the performance of MB-JPSRO improves. It emphasizes that the effectiveness of the model-based method relies on the quality of the trained environment model and also highlights the importance of accurate environment modeling in achieving robust offline equilibrium strategy computation.

## 6 CONCLUSION

We investigated the paradigm of offline equilibrium finding (Offline EF) in extensive-form games, aiming to compute equilibrium strategies from offline datasets. To address the lack of comprehensive datasets for evaluation, we first created offline EF datasets using established data collection methods. Next, we proposed a novel algorithm, **BOMB**, which combines behavior cloning with a model-based approach, enabling the adaptation of online equilibrium finding algorithms to the offline setting by a game model. To provide a deeper understanding, we conducted comprehensive theoretical and empirical analysis, offering performance guarantees of BOMB across different offline datasets. Finally, extensive experimental results validated the superiority of BOMB over existing offline RL algorithms, affirming its efficacy for computing equilibrium strategies in an offline manner. While this paper marks an important step toward offline learning in game theory, there remain limitations. Detailed discussions on limitations and future directions can be found in App. A. We hope our efforts inspire new avenues in equilibrium finding and accelerate research in large-scale game theory.

## ETHICS STATEMENT

This paper does not involve human subjects, sensitive personal data, or other ethical risks. The datasets used are synthetic, and no privacy or ethical concerns are associated with this study.

## REPRODUCIBILITY STATEMENT

We have made significant efforts to ensure the reproducibility of our work. The implementation of our proposed algorithms is submitted as anonymous supplementary materials, providing complete source code for reproduction of our experiments. The process of dataset generation is described in detail in the main paper, while the data structure and data analysis are provided in the appendix. Together, these materials ensure that the results presented in this paper can be reliably reproduced.

## STATEMENT ON THE USE OF LARGE LANGUAGE MODELS

We used ChatGPT to assist with the writing and polishing of this manuscript. The model was employed to improve grammar, clarity, and readability, but it did not contribute to the generation of research ideas, experimental design, implementation, or analysis. All technical content, including algorithms, proofs, and experimental results, was conceived and verified by the authors.

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

## A  FREQUENTLY ASKED QUESTIONS (FAQS)

**Q1: What are the potential impacts of this work?**

This work addresses a critical gap in offline learning within the domain of game theory. By introducing the offline learning framework, we demonstrate its potential to advance the application of game-theoretic methods to real-world problems and to stimulate new research directions in equilibrium computation. Notably, equilibrium strategies offer greater robustness than purely optimal strategies, particularly in security-sensitive and adversarial contexts. Consequently, the offline EF paradigm provides a valuable foundation for deriving robust strategies capable of effectively managing competitive and adversarial real-world challenges.

**Q2: Why is offline EF important and needed?**

Offline EF algorithms designed for adversarial environments is crucial in strictly competitive games, such as security games. This setting fundamentally differs from offline multi-agent reinforcement learning, which often emphasizes cooperation among agents rather than strict competition. Consider, for example, pursuit-evasion games, where the pursuer (defender) attempts to capture the evader (attacker). In such contexts, it is unrealistic to assume that the attacker follows a fixed strategy, since attackers are strategic and capable of adapting their behavior. If one were to apply a standard offline RL algorithm to learn the defender's optimal strategy solely from historical data, the resulting strategy could suffer significant utility loss, as it may be highly exploitable. Specifically, the attacker could deviate from past behaviors and instead adopt a best response to the defender's learned strategy. By contrast, computing a Nash Equilibrium (NE) provides a more robust solution, since equilibrium strategies are, by definition, non-exploitable.

**Q3: Is offline EF more difficult than offline RL or offline cooperative MARL?**

Traditional offline RL focuses on learning an optimal strategy for a single agent, with the goal of maximizing utility in a dynamic environment modeled as a Markov Decision Process (MDP). In this setting, the environment dynamics are independent of the actions of other agents. Offline cooperative multi-agent RL (MARL) extends this paradigm to multiple agents, but typically assumes that the agents work together toward a shared objective. Although the dynamics depend on the joint actions of all agents, the cooperative setting aligns their incentives, making it possible to optimize for collective performance without the concern of strategic exploitation.

By contrast, offline EF arises in strictly competitive games, where each player faces a strategic opponent. The effective dynamics for one player are determined jointly by the game structure and the opponent's strategy, and any change in the opponent's behavior alters the induced MDP. This makes the problem significantly more challenging than both offline RL and cooperative MARL, since a strategy learned from historical data may be highly exploitable if the opponent adapts. The offline EF framework addresses this difficulty by enabling the computation of Nash equilibrium (NE) strategies, which provide robustness against fully strategic and adversarial opponents.

**Q4: What are the differences between offline EF and Empirical Game-Theoretic Analysis?**

1) As described in (Wellman, 2006), Empirical Game-Theoretic Analysis (EGTA) requires access to a game simulator and conducts strategic reasoning through an iterative process of simulation and game-theoretic analysis. In this sense, the availability of a simulator is essential for EGTA. By contrast, the offline EF paradigm does not rely on a simulator. Instead, it operates under the offline learning setting, where only a fixed dataset of past interactions is available.

2) In EGTA, the empirical game is constructed from simulation outcomes obtained by executing a set of known strategies on the simulator. In contrast, in the offline EF paradigm, the offline dataset is generated using unknown strategy. While we use various behavior strategies to create multiple offline datasets, these strategies are not utilized directly within the offline EF paradigm. These distinctions highlight the fundamental differences between EGTA and offline EF.

**Q5: What are the novelties of the proposed offline EF algorithm – BOMB?**

To the best of our knowledge, we are the first to propose an empirical algorithm for computing equilibrium strategies from offline datasets, i.e., the offline EF paradigm. Unlike traditional offline RL algorithms, which are typically categorized as either model-based or model-free, our algorithm combines the advantages of both to efficiently compute equilibrium strategies in the offline setting. The

proposed BOMB framework integrates behavior cloning (BC) with a model-based (MB) component, supported by novel parameter estimation techniques. By incorporating an environment model into the MB component, we enable the adaptation of existing online equilibrium finding algorithms for use in offline settings. Furthermore, we design multiple methods for selecting the combination parameter, providing flexibility to accommodate different scenarios depending on whether limited online interaction is available. Experimental results show that neither the BC nor the MB approach consistently achieves strong performance in isolation, whereas BOMB outperforms both across all tested cases. These results demonstrate that the BOMB framework successfully unifies the strengths of BC and MB, yielding a highly effective solution for offline equilibrium computation.

**Q6: What are the limitations and future works for this paper?**

This work has several limitations that we aim to address in the future. First, the games considered in this study are relatively small-scale. Expanding to large-scale games, such as Texas Hold'em poker (Brown & Sandholm, 2018) and football games (Liu et al., 2022), will be a focus of future work. Second, this work primarily targets NE and CCE as solution conceptions. In future research, we plan to incorporate additional solution concepts, such as quantal response equilibrium (QRE) (McKelvey & Palfrey, 1995) and $\alpha$-rank (Omidshafiei et al., 2019). Investigating the generalizability of both the datasets and the BOMB framework to these novel solution concepts will further broaden the scope of offline EF. Third, the relationships between the datasets and offline EF algorithms require further exploration. Instead of relying on datasets curated by researchers, we aim to apply offline EF to human-play datasets, moving toward real-world deployment (Wang et al., 2024). This will help bridge the gap between theoretical research and practical applications.

# B    RELATED WORK

## B.1    EQUILIBRIUM FINDING ALGORITHMS.

Contemporary state-of-the-art algorithms for solving IIEFGs can be broadly categorized into two groups: no-regret methods derived from CFR (Zinkevich et al., 2007), and incremental strategy-space generation methods of the PSRO framework (Lanctot et al., 2017).

For the first group, CFR is a family of iterative methods for approximately solving IIEFGs. Let $\sigma_i^t$ be the strategy used by player $i$ in iteration $t$. We use $u_i(\sigma, h)$ to define the expected utility of player $i$ given that the history $h$ is reached and all players act according to strategy $\sigma$ from that point on. Accordingly, $u_i(\sigma, h \cdot a)$ is used to define the expected utility of player $i$ given that the history $h$ is reached and all players play according to strategy $\sigma$ except player $i$ selects action $a$ in history $h$. Formally, $u_i(\sigma, h) = \sum_{z \in Z} \pi^\sigma(h, z) u_i(z)$ and $u_i(\sigma, h \cdot a) = \sum_{z \in Z} \pi^\sigma(h \cdot a, z) u_i(z)$. The *counterfactual value* of the information set $I$, $v_i^\sigma(I)$, is the expected value of information set $I$ given that player $i$ attempts to reach it. This value is the weighted average of the expected utility of each history in the information set. The weight is proportional to the contribution of all players except player $i$ to reach each history. Thus, $v_i^\sigma(I) = \sum_{h \in I} \pi_{-i}^\sigma(h) u_i(\sigma, h)$. For any action $a \in A(I)$, the counterfactual value of action $a$ is $v_i^\sigma(I, a) = \sum_{h \in I} \pi_{-i}^\sigma(h) u_i(\sigma, h \cdot a)$. The *instantaneous counterfactual regret* for an action $a$ in information set $I$ during iteration $t$ is $r^t(I, a) = v_{P(I)}^{\sigma^t}(I, a) - v_{P(I)}^{\sigma^t}(I)$. Therefore, the conterfactual regret for an action $a$ in inforamtion set $I$ on iteration $T$ is $R^T(I, a) = \sum_{t=1}^T r^t(I, a)$. In vanilla CFR, players use *Regret Matching* to pick a distribution over available actions in an information set proportional to the cumulative regret of those actions. Formally, in iteration $T + 1$, player $i$ selects action $a \in A(I)$ according to probabilities

$$\sigma^{T+1}(I, a) = \begin{cases} \frac{R_+^T(I,a)}{\sum_{b \in A(I)} R_+^T(I,b)} & \text{if } \sum_{b \in A(I)} R_+^T(I, b) > 0, \\ \frac{1}{|A(I)|} & \text{otherwise,} \end{cases}$$

where $R_+^T(I, a) = \max\{R^T(I, a), 0\}$ is the position portion of the regret value since we often are most concerned about the cumulative regret when it is positive. If a player acts according to regret matching in the information set $I$ on every iteration, then in iteration $T$, $R^T(I) \le \Delta_i \sqrt{|A_i|} \sqrt{T}$ where $\Delta_i = \max_z u_i(z) - \min_z u_i(z)$ is the range of utilities of player $i$. Moreover, $R_i^T \le \sum_{I \in \mathcal{I}_i} R^T(I) \le |\mathcal{I}_i| \Delta_i \sqrt{|A_i|} \sqrt{T}$. Therefore, $\lim_{T \to \infty} \frac{R_i^T}{T} = 0$. In two-player zero-sum games,

if both players' average regret $\frac{R_i^T}{T} \leq \epsilon$, their average strategies $(\overline{\sigma}_1^T, \overline{\sigma}_2^T)$ over all iterations form a $2\epsilon$-equilibrium (Waugh et al., 2009). Some CFR-based variants are proposed to solve large-scale imperfect-information extensive-form games. Some sampling-based CFR variants (Lanctot et al., 2009; Gibson et al., 2012; Schmid et al., 2019) are proposed to effectively solve large-scale games by traversing a subset of the game tree instead of the whole game tree. With the development of deep learning techniques, neural network function approximation can be applied to the CFR algorithm. Deep CFR (Brown et al., 2019), Single Deep CFR (Steinberger, 2019), and Double Neural CFR (Li et al., 2019) are algorithms using deep neural networks to replace the tabular representation.

For the second group, PSRO is a general framework that scales Double Oracle (DO) (McMahan et al., 2003) to large extensive-form games via using RL to compute the best response approximately. To make PSRO more effective in solving large-scale games, Pipeline PSRO (P2SRO) (McAleer et al., 2020) is proposed by parallelizing PSRO with convergence guarantees. Extensive-Form Double Oracle (XDO) (McAleer et al., 2021) is a version of PSRO where the restricted game allows mixing population strategies not only at the root of the game but every information set. It can guarantee to converge to an approximate NE in a number of iterations that are linear in the number of information sets, while PSRO may require a number of iterations exponential in the number of information sets. Neural XDO (NXDO), as a neural version of XDO, learns approximate best response strategies through any deep RL algorithm. Recently, Anytime Double Oracle (ADO) (McAleer et al., 2022), a tabular double oracle algorithm for two-player zero-sum games, was proposed to converge to an NE while decreasing exploitability from one iteration to the next. Anytime PSRO (APSRO) as a version of ADO calculates best responses via RL algorithms. Except for NEs, we also consider (Coarse) Correlated equilibrium ((C)CE). Joint Policy Space Response Oracles (JPSRO) (Marris et al., 2021) is proposed for training agents in n-player, general-sum extensive-form games, which provably converges to (C)CEs. The excellent performance of these equilibrium-finding algorithms depends on the interactions with the actual game environment or a precise simulator. Therefore, these algorithms cannot directly be applied to the offline EF paradigm.

### B.2 OPPONENT MODELING.

Opponent modeling algorithm is necessary for multi-agent settings where secondary agents with competing goals also adapt their strategies, yet it remains challenging because policies interact with each other and change (He et al., 2016). One simple idea of opponent modeling is to build a model each time a new opponent or group of opponents is encountered (Zheng et al., 2018). However, it is infeasible to learn a model every time. A better approach is to represent an opponent's policy with an embedding vector. Grover et al. (2018) use a neural network as an encoder, taking the trajectory of one agent as input. Imitation learning and contrastive learning are also used to train the encoder. Then, the learned encoder can be combined with reinforcement learning algorithms by feeding the generated representation into the policy and/or value network. DRON (He et al., 2016) and DPIQN (Hong et al., 2017) are two algorithms based on DQN, which use a secondary network that takes observations as input and predicts opponents' actions. However, if the opponents can also learn, these methods become unstable. Therefore, it is necessary to take the learning process of opponents into account. Foerster et al. (2017) propose a method named Learning with Opponent-Learning Awareness (LOLA), in which each agent shapes the anticipated learning of the other agents in the environment. Further, the opponents may still be learning continuously during execution. Therefore, Al-Shedivat et al. (2017) propose a method based on a meta-policy gradient named Mata-MPG. It uses trajectories from current opponents to perform multiple meta-gradient steps and constructs a policy that favors updating the opponents. Meta-MAPG (Kim et al., 2021) extends Mate-MPG by including an additional term that accounts for the impact of the agent's current policy on the future policies of opponents, similar to LOLA. Yu et al. (2021b) propose model-based opponent modeling (MBOM), which employs the environment model to adapt to various opponents. In our offline EF paradigm, our goal is to compute the equilibrium strategy based on the offline dataset. Applying opponent modeling is not enough for the offline EF paradigm since it only aims at computing the best response strategy instead of the equilibrium strategy.

### B.3 EMPIRICAL GAME THEORETIC ANALYSIS.

Empirical game theoretic analysis (EGTA) is an empirical methodology that bridges the gap between game theory and simulation for practical strategic reasoning (Wellman, 2006). In EGTA, game mod-

els are iteratively extended through a process of generating new strategies based on learning from experience with prior strategies. The strategy exploration problem (Jordan et al., 2010) that how to efficiently assemble an efficient portfolio of policies for EGTA is the most challenging problem. Schvartzman & Wellman (2009b) deploy tabular RL as a best-response oracle in EGTA for strategy generation. They also build the general problem of strategy exploration in EGTA and investigate whether better options exist beyond best-responding to an equilibrium (Schvartzman & Wellman, 2009a). Investigation of strategy exploration was advanced significantly by the introduction of the Policy Space Response Oracle (PSRO) framework (Lanctot et al., 2017) which is a flexible framework for iterative EGTA, where at each iteration, new strategies are generated through reinforcement learning. Note that when employing NE as the meta-strategy solver, PSRO reduces to the double oracle (DO) algorithm (McMahan et al., 2003). In EGTA, a space of strategies is examined through simulation, which means that it needs a simulator, and the policies are known in advance. However, in the offline EF paradigm, only an offline dataset is provided. Therefore, techniques in EGTA cannot be directly applied to the offline EF paradigm.

### B.4    OFFLINE REINFORCEMENT LEARNING.

Offline reinforcement learning (offline RL) is a *data-driven* paradigm that learns exclusively from static datasets of previously collected interactions, making it feasible to extract policies from large and diverse training datasets (Levine et al., 2020). This paradigm can be extremely valuable in settings where online interaction is impractical, either because data collection is expensive or dangerous (e.g., in robotics (Singh et al., 2021), education (Singla et al., 2021), healthcare (Liu et al., 2020), and autonomous driving (Kiran et al., 2022)). Therefore, efficient offline RL algorithms have a much broader range of applications than online RL and are particularly appealing for real-world applications (Prudencio et al., 2022). Due to its attractive characteristics, there have been a lot of recent studies. Here, we can divide the research of offline RL into two categories: model-based algorithm and model-free algorithm.

Model-free offline RL algorithms learn a good policy directly from the offline dataset. To do this, there are two types of algorithms: actor-critic and imitation learning methods. Those actor-critic algorithms focus on implementing policy regularization and value regularization based on existing reinforcement learning algorithms. Haarnoja et al. (2018) propose soft actor-critic (SAC) by adding an entropy regularization term to the policy gradient objective. This work mainly focuses on policy regularization. For the research of value regularization, an offline RL method named Constrained Q-Learning (CQL) (Kumar et al., 2020) learns a lower bound of the true Q-function by adding value regularization terms to its objective. Another line of model-free offline RL research is imitation learning which mimics the behavior policy based on the offline dataset. Chen et al. (2020) propose a method named Best-Action Imitation Learning (BAIL), which fits a value function, then uses it to select the best actions. Meanwhile, Siegel et al. (2020) propose a method that learns an Advantage-weighted Behavior Model (ABM) and uses it as a prior in performing Maximum a-posteriori Policy Optimization (MPO) (Abdolmaleki et al., 2018). It consists of multiple iterations of policy evaluation and prior learning until they finally perform a policy improvement step using their learned prior to extracting the best possible policy.

Model-based algorithms rely on the offline dataset to learn a dynamics model or a trajectory distribution used for planning. The trajectory distribution induced by models is used to determine the best set of actions to take at each given time step. Kidambi et al. (2020) propose a method named Model-based Offline Reinforcement Learning (MOReL), which measures their model's epistemic uncertainty through an ensemble of dynamics models. Meanwhile, Yu et al. (2020) propose another method named Model-based Offline Policy Optimization (MOPO), which uses the maximum prediction uncertainty from an ensemble of models. Concurrently, Matsushima et al. (2020) propose the BehaviorREgularized Model-ENsemble (BREMEN) method, which learns an ensemble of models of the behavior MDP, as opposed to a pessimistic MDP. In addition, it implicitly constrains the policy to be close to the behavior policy through trust-region policy updates. More recently, Yu et al. (2021a) proposed a method named Conservative Offline Model-Based policy Optimization (COMBO), a model-based version of CQL. The main advantage of COMBO concerning MOReL and MOPO is that it removes the need for uncertainty quantification in model-based offline RL approaches, which is challenging and often unreliable. However, the above offline RL algorithms cannot be directly applied to the offline EF paradigm, which we have described in Section 2 and experimental results also empirically verify this claim.

# C   DATASETS

## C.1   DATASET FORMAT

To clarify the structure of the offline EF dataset, we provide an illustrative example. As introduced in the main paper, each data point is represented as $(s_t, a_t, s_{t+1}, u_{t+1}, d_{t+1})$, where $s_t = (I_1^t, I_2^t, ..., I_n^t, GI, p^t, A(I_{p^t}^t))$. Here, $I_1^t$ denotes the information set of player $i$ at time step $t$, $GI$ denotes the game information, $p^t$ is the player acting at time step $t$, and $A(I_{p^t}^t)$ is the set of available actions at information set $I_1^t$. Notably, if $s_{t+1}$ is a terminal state, i.e., $d_{t+1} = 1$, we set $p^{t+1} = -1$ to indicate that no player is required to make a decision at this state. Fig. 12 presents an example of a two-player IIEFG $\mathcal{G}$, where $I_1$ and $I_2$ denote the in-

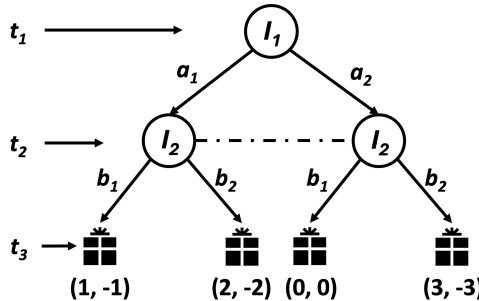

Figure 12: An example game.

formation sets for Player 1 and Player 2, respectively. If the offline dataset $\mathcal{D}$ provides full coverage of all state-action pairs, it would contain the following data points:

$$((I_1^{t_1} = I_1, I_2^{t_1} = \emptyset, \emptyset, 1, \{a_1, a_2\}), a_1, (I_1^{t_2} = I_1 a_1, I_2^{t_2} = I_2, \emptyset, 2, \{b_1, b_2\}), (0,0), 0),$$

$$((I_1^{t_1} = I_1, I_2^{t_1} = \emptyset, \emptyset, 1, \{a_1, a_2\}), a_2, (I_1^{t_2} = I_1 a_2, I_2^{t_2} = I_2, \emptyset, 2, \{b_1, b_2\}), (0,0), 0),$$

$$((I_1^{t_2} = I_1 a_1, I_2^{t_2} = I_2, \emptyset, 2, \{b_1, b_2\}), b_1, (I_1^{t_3} = I_1 a_1, I_2^{t_3} = I_2 b_1, \emptyset, -1, \emptyset), (1,-1), 1),$$

$$((I_1^{t_2} = I_1 a_1, I_2^{t_2} = I_2, \emptyset, 2, \{b_1, b_2\}), b_2, (I_1^{t_3} = I_1 a_1, I_2^{t_3} = I_2 b_2, \emptyset, -1, \emptyset), (2,-2), 1),$$

$$((I_1^{t_2} = I_1 a_2, I_2^{t_2} = I_2, \emptyset, 2, \{b_1, b_2\}), b_1, (I_1^{t_3} = I_1 a_2, I_2^{t_3} = I_2 b_1, \emptyset, -1, \emptyset), (0,0), 1),$$

$$((I_1^{t_2} = I_1 a_2, I_2^{t_2} = I_2, \emptyset, 2, \{b_1, b_2\}), b_2, (I_1^{t_3} = I_1 a_2, I_2^{t_3} = I_2 b_2, \emptyset, -1, \emptyset), (3,-3), 1).$$

Note that in states $(I_1^{t_2} = I_1 a_1, I_2^{t_2} = I_2, \emptyset, 2, \{b_1, b_2\})$ and $(I_1^{t_2} = I_1 a_2, I_2^{t_2} = I_2, \emptyset, 2, \{b_1, b_2\})$, the information set for Player 2, $I_2^{t_2}$, is identical, as shown in Fig. 12. However, because our dataset is constructed from the perspective of the entire game, these states can still be distinguished by incorporating the game information of other players together with the game information $GI$. In this specific example, there is no chance node, so $GI$ is empty. If the game were to include a chance node, its outcomes would be recorded in $GI$. This additional information allow us to differentiate game states that may otherwise appear indistinguishable from the perspective of a single player. In this way, dataset ensures sufficient granularity to uniquely identify all relevant states in the game.

## C.2   VISUALIZATION

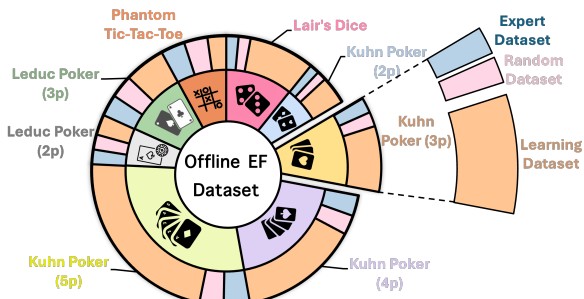

Figure 13: Visualization of the offline EF dataset

Fig. 13 provides a comprehensive view of our offline EF dataset, which includes data collected for eight games: two-player Kuhn poker, three-player Kuhn poker, four-player Kuhn poker, five-player Kuhn poker, two-player Leduc poker, three-player Leduc poker, Phantom Tic-Tac-Toe, and Liar's

Dice. For each game, we generated three types of datasets, a random dataset, an expert datset, and a learning dataset, following our data collection methods. To validate the diversity of these collected offline datasets and gain insights into them, we developed a visualization method for comparison. First, we generate the game tree for each game. Subsequently, we traverse the game tree using depth-first search (DFS) (Tarjan, 1972) and assign an index to each leaf node based on the DFS results. Then, we count the frequency of each leaf node within the dataset. The reason why we do this is that each leaf node represents a unique sampled trajectory originating from the root node of the game tree. As a result, the frequency of leaf nodes can effectively capture the distribution of the dataset. Finally, these frequency data can be plotted to visualize. Fig. 14 shows the visualized distributions for some datasets. From these figures, we can find that in the random dataset, the frequency of leaf nodes is nearly uniform, whereas, in the expert dataset, the frequency of leaf nodes is uneven. The distribution of the learning dataset and the hybrid dataset falls between that of the expert dataset and the random dataset. These observations confirm that the distribution of these datasets differs, thus validating the diversity of our proposed offline datasets.

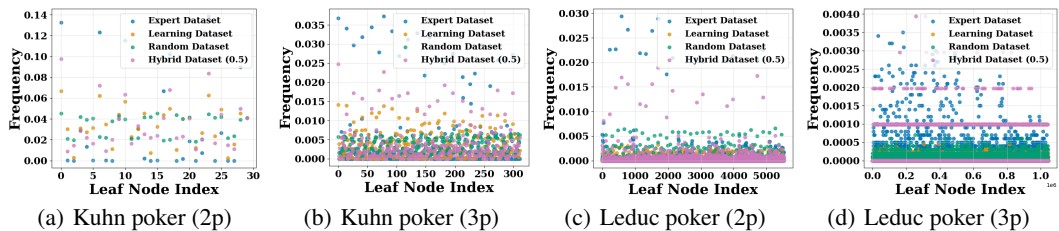

|  (a) Kuhn poker (2p) | (b) Kuhn poker (3p) | (c) Leduc poker (2p) | (d) Leduc poker (3p) |

Figure 14: Frequency of leaf node in different offline datasets.

## D    THEORETICAL ANALYSIS

In this section, we provide a comprehensive theoretical analysis of the offline EF paradigm and our BOMB framework to facilitate the understanding of the offline EF paradigm and BOMB framework. We first provide the minimal dataset assumption that is sufficient to compute the equilibrium strategy in the offline setting. Then we provide a general generalization bound for training neural network models. Finally, we give the performance guarantee for our algorithm. In the following sections, we assume that all extensive-form games discussed here are perfect recall and timetable.

### D.1    MINIMAL DATASET ASSUMPTION FOR OFFLINE EF

As demonstrated in offline RL papers (Rashidinejad et al., 2021; Xie et al., 2021), a dataset coverage condition over the optimal policy is sufficient for offline learning. Therefore, it is straightforward to extend this dataset coverage assumption to the offline EF paradigm. In the main paper, we have proved that the dataset generated by the equilibrium strategy is not sufficient for computing the equilibrium strategy in an offline manner by providing a counter-example. Furthermore, we also provide another dataset assumption related to the equilibrium strategy, shown in the following assumption.

**Assumption D.1.** (Single Strategy Coverage) The offline dataset $\mathcal{D}$ is said to be a *single strategy coverage* if the equilibrium strategy profile $\sigma^*$ is covered by the offline dataset $\mathcal{D}$, i.e., for each player $i$, each information set $I_i$, and action $a_i$ with $\sigma_i^*(I_i, a_i) > 0$, there is a corresponding state-action pair $(s_t, a_i)$ in $D$.

Subsequently, a question arises: *is the single strategy coverage assumption also sufficient for computing equilibrium strategy in the offline setting?* We employ the following theorem to answer this question and elucidate the rationale behind this.

**Theorem D.2.** *Single strategy coverage assumption over offline dataset $\mathcal{D}$ is not sufficient for computing computing an $\epsilon$-equilibrium for an arbitrarily small $\epsilon$ in the offline setting.*

*Proof.* We prove this theorem by providing a counterexample. First, we consider two two-player IIEFGs $G_1$ and $G_2$, represented in Fig. 15. We can easily find that the NE of the game $G_1$ is

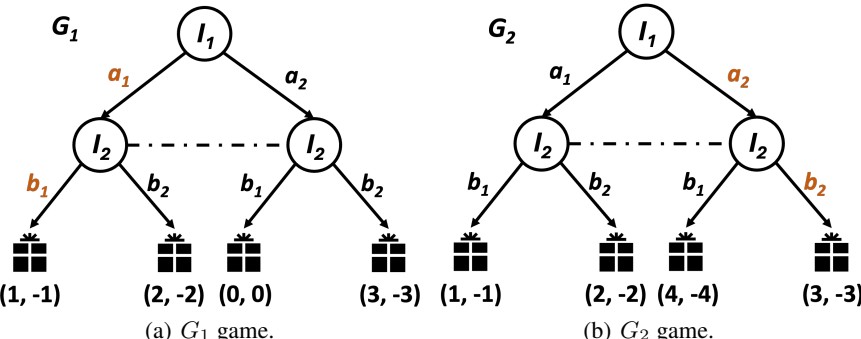

Figure 15: Counterexample for proving Theorem D.2.

strategy profile $\sigma^1 = (\sigma_1^1, \sigma_2^1) = (\{I_1 : a_1\}, \{I_2 : b_1\})$, i.e., Player 1 plays $a_1$ at information set $I_1$ and Player 2 plays $b_1$ at information set $I_2$. The NE of the game $G_2$ is strategy profile $\sigma^2 = (\sigma_1^2, \sigma_2^2) = (\{I_1 : a_2\}, \{I_2 : b_2\})$. Next, we consider an offline dataset $\mathcal{D}$ which is generated using a strategy profile $\sigma_{\mathcal{D}}$. The $\sigma_{\mathcal{D}}$ is set to be the uniform distribution over the strategy profiles $\sigma^1$ and $\sigma^2$, which means that the dataset $\mathcal{D}$ covers both $\sigma^1$ and $\sigma^2$. Therefore, the offline dataset $\mathcal{D}$ satisfies the single strategy coverage assumption for these two games $G_1$ and $G_2$. However, no algorithm can distinguish these two games only based on the dataset $\mathcal{D}$ since these two games are both consistent on the dataset $\mathcal{D}$. In conclusion, the single strategy coverage assumption is not sufficient for computing an $\epsilon$-equilibrium for an arbitrarily small $\epsilon$ in the offline setting. □

From the above proof, we understand that the single strategy coverage assumption is sufficient for computing the optimal strategy in the offline RL setting, but is inadequate for computing an NE strategy in the offline EF setting. The intuition behind this difference lies in the nature of the two problems: in offline RL, data from only two actions is sufficient to determine which action is better; however, in offline EF, data from only two action pairs cannot determine which action pair is closer to NE. Identifying an NE strategy requires referencing additional action pairs for comparison and inference. Based on this analysis, Cui & Du (2022) provided a minimal coverage assumption that is sufficient for computing NE in two-player zero-sum Markov games, which is defined as follows,

**Assumption D.3.** (Deterministic Unilateral Coverage) For all deterministic strategy $\sigma_i$ for player $i$, $(\sigma_i, \sigma_{-i}^*)$ are covered by the dataset, where $(\sigma_1^*, ..., \sigma_n^*)$ is one NE strategy.

**Assumption D.4.** (Unilateral Coverage) For all (possible stochastic) strategy $\sigma_i$ for all player $i$, $(\sigma_i, \sigma_{-i}^*)$ are covered by the dataset, where $(\sigma_1^*, ..., \sigma_n^*)$ is one NE strategy.

Note that the deterministic unilateral coverage assumption is equivalent to the unilateral coverage assumption. The intuition behind this is that any mixed strategy can be represented by a combination of deterministic strategies. Therefore, if all deterministic strategies are covered by the dataset, then all mixed strategies are also covered. Based on this finding, in the following proof, we only consider all deterministic strategies. Previously, Cui & Du (2022) established that the unilateral coverage assumption is the minimal sufficient condition for computing an NE strategy in the two-player zero-sum Markov games. However, this unilateral coverage assumption over the offline dataset is **not sufficient** for our model-based method to compute the equilibrium strategy in the offline setting. We formally proved this limitation through the following theorem.

**Theorem D.5.** *The unilateral coverage assumption over the offline dataset $\mathcal{D}$ is not sufficient for our model-based method to converge to an $\epsilon$-equilibrium for an arbitrarily small $\epsilon$ in the offline setting.*

*Proof.* We prove this theorem by providing a counterexample. First, we consider an IIEFG $M_3$, represented in Fig. 16(a). We can easily find that the NE strategy of game $G_3$ is strategy profile $\sigma^* = (\sigma_1, \sigma_2) = (\{I_1 : a_1\}, \{I_2 : b_1\})$. To build a dataset $\mathcal{D}$ satisfying the unilateral coverage assumption, the dataset needs to cover $(\sigma_1^*, \sigma_2)$ for all deterministic strategies $\sigma_2$ and $(\sigma_1, \sigma_2^*)$ for all deterministic strategies $\sigma_1$. We show the state-action pairs covered by these strategy profiles in Figs. 16(b)-16(c). It means that if the dataset $\mathcal{D}$ satisfies the unilateral coverage assumption, then the dataset $\mathcal{D}$ would cover these state-action pairs marked by these orange lines. When applying

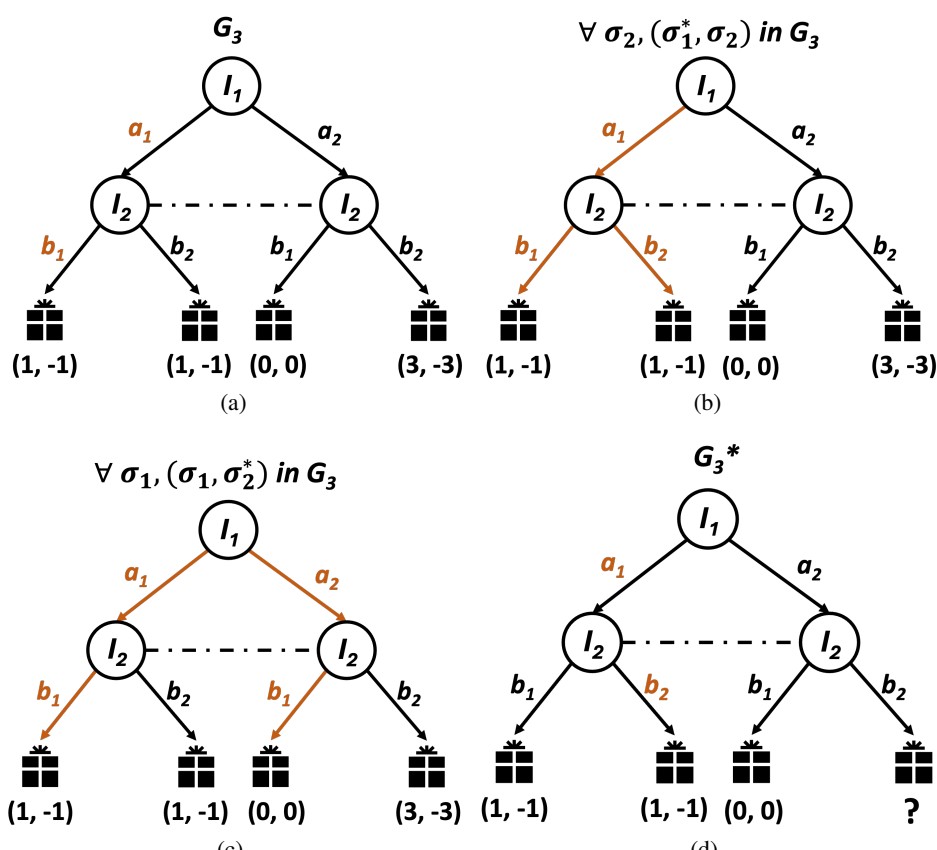

Figure 16: Counter-example for proving Theorem D.5.

our model-based method to the dataset $\mathcal{D}$, the first step is to train an environment model using the dataset $\mathcal{D}$. Assume that the environment model can be trained well (i.e., Assumption 4.3 holds), which means that the environment model can precisely represent all game information in the dataset. Therefore, the game represented by the trained environment model would be $G_3^*$ in Fig. 16(d). Note that there is some missing data in the game. Although our trained environment model can give approximate results for these missing data, it may result in a different equilibrium strategy. For example, if the missing value in $G_3^*$ is $(0, 0)$ or $(-1, 1)$, then the strategy profile $\sigma = (\sigma_1, \sigma_2') = (\{I_1 : a_1\}, \{I_2 : b_2\})$ would be the NE strategy of game $G_3^*$. However, the strategy profile $\sigma$ is not the NE strategy for the original game $G_3$. Therefore, the unilateral coverage assumption is not sufficient for our model-based method to converge to an $\epsilon$-equilibrium for an arbitrarily small $\epsilon$. $\square$

To ensure the convergence of our model-based method, we provide a minimal dataset coverage assumption that guarantees our model-based method converges to the equilibrium strategy of the original game under the offline setting.

**Definition D.6** (Definition 4.1). An offline dataset $\mathcal{D}$ is said to be a *uniform coverage* of an IIEFG $\mathcal{G}$ if and only if it covers all possible state-action pairs in the game. Formally, it covers $(s_t, a_t, s_{t+1}, u_{t+1}, d_{t+1}), \forall s_t, a_t \in A(s_t)$ and $s_{t+1} \in T(s_t, a_t)$ where $T$ is the transition function.

**Theorem D.7** (Theorem 4.4). *Let $\sigma_{MB(\mathcal{D})}$ be the strategy learned by **MB algorithm** using the offline dataset $\mathcal{D}$ with sufficient data under Assump. 4.3. Then, $\sigma_{MB(\mathcal{D})}$ is guaranteed to be an $\epsilon$-equilibrium strategy of IIEFG $\mathcal{G}$ if $\mathcal{D}$ is a uniform coverage of $\mathcal{G}$ and $\sigma_{MB(\mathcal{D})}$ is an $\epsilon$-equilibrium strategy of the trained environment used in the MB algorithm. If either condition fails, the guarantee may no longer hold.*

*Proof.* From the example in the proof of Theorem D.5, we find that a slight violation of the uniform coverage assumption, i.e., only one state-action pair is missing, will impede the computation of the

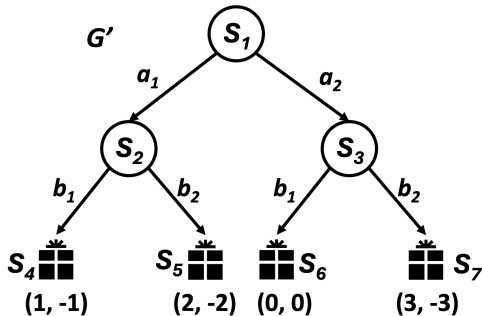

Figure 17: $G'$ Game.

equilibrium strategy using our model-based method. In other words, any state-action pair that is not covered by the dataset may cause failure in computing the equilibrium strategy of the original game using our model-based method.

Next, we need to prove that the dataset satisfying the uniform coverage assumption can guarantee the convergence to the equilibrium strategy of the original game using our model-based method. In our model-based method, we need to train an environment model based on the offline dataset. Therefore, to prove the convergence guarantee under the uniform coverage dataset assumption, we need to verify whether the game reconstructed from the dataset satisfying the uniform coverage assumption is the same as the original game. Here, we reuse the example in the App. C.1. In that example, the offline datset $\mathcal{D}$ of the IIEFG $\mathcal{G}$ covers all state-action pairs. Therefore, the offline dataset $\mathcal{D}$ satisfies the uniform coverage dataset assumption. From the offline dataset $\mathcal{D}$, we can easily rebuild the game $G'$, as shown in Fig. 17. In the game $G'$,

$$S_1 = (I_1^{t_1} = I_1, I_2^{t_1} = \emptyset, \emptyset, 1, \{a_1, a_2\}), S_2 = (I_1^{t_2} = I_1a_1, I_2^{t_2} = I_2, \emptyset, 2, \{b_1, b_2\}),$$

$$S_3 = (I_1^{t_2} = I_1a_2, I_2^{t_2} = I_2, \emptyset, 2, \{b_1, b_2\}), S_4 = (I_1^{t_3} = I_1a_1, I_2^{t_3} = I_2b_1, \emptyset, -1, \emptyset),$$

$$S_5 = (I_1^{t_3} = I_1a_1, I_2^{t_3} = I_2b_2, \emptyset, -1, \emptyset), S_6 = (I_1^{t_3} = I_1a_2, I_2^{t_3} = I_2b_1, \emptyset, -1, \emptyset),$$

$$S_7 = (I_1^{t_3} = I_1a_2, I_2^{t_3} = I_2b_2, \emptyset, -1, \emptyset).$$

Especially, for game states $S_2$ and $S_3$, the player acting is both Player 2, and the information set for Player 2 is the same. Therefore, these two game states correspond to different game nodes under the same information set. Although Player 2 cannot distinguish these two game states, from the perspective of the game, we can still distinguish them by the information set of Player 1. Particularly, if there is a chance node in the game, the result of the chance node would be recorded in $GI$ within the game state $S$. Therefore, we can still distinguish these game states by game information $GI$. Since the dataset satisfying the uniform coverage assumption covers all state-action pairs, the links between game states can be built following these data points in the dataset. According to Assumption 4.3, the error in training the environment game model based on $\mathcal{D}$ can be considered negligible. Consequently, the trained environment game model is identical to the original game $\mathcal{G}$, as the dataset $\mathcal{D}$ provides full coverage of all state transitions. Therefore, we can find that the reconstructed game tree has the same game states and the same transition function as the original game, thereby the same equilibrium strategy. Therefore, our reconstructed game model can provide the same information as the underlying game of the offline dataset. Then applying our model-based equilibrium finding algorithm to the reconstructed game model definitely can converge to the equilibrium strategy of the underlying game in the offline setting. Formally, if $\sigma_{MB(\mathcal{D})}$ is an $\epsilon$-equilibrium strategy for the trained environment game model, it is also an $\epsilon$-equilibrium strategy for the original game $\mathcal{G}$. □

So far, we have proved that the uniform dataset coverage assumption is sufficient for our model-based method to converge to the equilibrium strategy under the offline setting. For our behavior cloning method, these dataset coverage assumptions may not be sufficient to converge to the equilibrium strategy since its performance mainly depends on the underlying behavior strategy of the dataset. In the following theorem, we provide a minimal dataset coverage assumption for our behavior cloning method to converge to the equilibrium strategy in the offline setting.

**Definition D.8** (Definition 4.2). An offline dataset $\mathcal{D}$ is said to be an $\epsilon$-**equilibrium coverage** over an IIEFG $\mathcal{G}$ if and only if its underlying behavior strategy $\sigma_{\mathcal{D}}$ satisfies: $\mathrm{GAP}(\sigma_{\mathcal{D}}, \sigma^*) < \epsilon$, where $\sigma_{\mathcal{D}}$

is defined as: $\sigma_\mathcal{D}(s_t, a_t) = \frac{C(s_t, a_t)}{C(s_t)}$ and $\sigma_D(s_t, a_t) > 0$ for all $s_t$ and $a_t \in A(s_t)$, with $C(s_t, a_t)$ and $C(s_t)$ denoting the counts of occurrences of state-action pair $(s_t, a_t)$ and the state $s_t$ in $\mathcal{D}$, respectively.

This definition ensures that the unique correspondence relationship between the equilibrium-covered dataset and the equilibrium strategy. Specifically, the dataset is generated by the equilibrium strategy and the strategy represented by the dataset would be the same as the equilibrium strategy.

**Theorem D.9** (Theorem 4.5). *Let $\sigma_{BC(\mathcal{D})}$ be the strategy learned by **BC algorithm** using the offline dataset $\mathcal{D}$ with sufficient data under Assump. 4.3. Then, $\sigma_{BC(\mathcal{D})}$ is guaranteed to be an $\epsilon$-equilibrium strategy of IIEFG $\mathcal{G}$ if $\mathcal{D}$ is an $\epsilon$-equilibrium coverage of $\mathcal{G}$. Otherwise, the guarantee may no longer hold.*

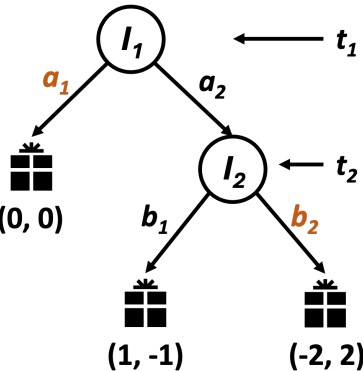

Figure 18: Game example.

*Proof.* According to Assumption 4.3, the error in training the behavior cloning strategy $\sigma_{BC(\mathcal{D})}$ from the dataset $\mathcal{D}$ is negligible. Therefore, by the behavior cloning process, $\sigma_{BC(\mathcal{D})}$ is identical to the behavior strategy underlying $\mathcal{D}$, i.e., $\sigma_{BC(\mathcal{D})} = \sigma_\mathcal{D}$. Consequently, if $\mathcal{D}$ is an $\epsilon$-equilibrium coverage of $\mathcal{G}$, then $\sigma_{BC(\mathcal{D})}$ is an $\epsilon$-equilibrium strategy for the IIEFG $\mathcal{G}$, and vice visa, as $\mathrm{GAP}(\sigma_\mathcal{D}, \sigma^*) < \epsilon$ if and only if $\mathrm{GAP}(\sigma_{BC(\mathcal{D})}, \sigma^*) < \epsilon$. Next, we prove that any slight violation of these conditions would invalidate the convergence result.

Here, we reuse the example in Section 4.2, as shown in Fig. 18. Note that the NE strategy of the game is a pure strategy, i.e, $\sigma^* = (\sigma_1^*, \sigma_2^*) = (\{I_1 : a_1\}, \{I_2 : b_2\})$. If we use this equilibrium strategy to generate the offline dataset $\mathcal{D}$, then $\mathcal{D}$ would only include the data point $((I_1^{t_1} = I_1, I_2^{t_1} = \emptyset, \emptyset, 1, \{a_1, a_2\}), a_1, (I_1^{t_2} = I_1 a_1, I_2^{t_2} = \emptyset, \emptyset, -1, \emptyset), (0, 0), 1)$. We cannot get the equilibrium strategy only from $\mathcal{D}$. In this example game, the offline dataset $\mathcal{D}$ is generated by a pure equilibrium strategy instead of a fully mixed equilibrium strategy, and the behavior cloning method cannot get the equilibrium strategy from the offline dataset $\mathcal{D}$ since there is no information about Player 2. Another example is the dataset $D'$ covering the equilibrium strategy $\sigma^*$, i.e., the offline dataset $D'$ includes the following data points:

$$((I_1^{t_1} = I_1, I_2^{t_1} = \emptyset, \emptyset, 1, \{a_1, a_2\}), a_1, (I_1^{t_2} = I_1 a_1, I_2^{t_2} = \emptyset, \emptyset, -1, \emptyset), (0, 0), 1),$$

$$((I_1^{t_1} = I_1, I_2^{t_1} = \emptyset, \emptyset, 1, \{a_1, a_2\}), a_2, (I_1^{t_2} = I_1 a_2, I_2^{t_2} = I_2, \emptyset, 2, \{b_1, b_2\}), (0, 0), 0),$$

$$((I_1^{t_2} = I_1 a_2, I_2^{t_2} = I_2, \emptyset, 2, \{b_1, b_2\}), b_2, (I_1^{t_2} = I_1 a_2, I_2^{t_2} = I_2 b_2, \emptyset, -1, \emptyset), (-2, 2), 1).$$

Data point $((I_1^{t_1} = I_1, I_2^{t_1} = \emptyset, \emptyset, 1, \{a_1, a_2\}), a_2, (I_1^{t_2} = I_1 a_2, I_2^{t_2} = I_2, \emptyset, 2, \{b_1, b_2\}), (0, 0), 0)$ should also be visited to cover the equilibrium strategy of Player 2. Although $D'$ covers the equilibrium strategy, $D'$ does not satisfy the $\epsilon$-equilibrium coverage assumption since $D'$ is not created by the $\epsilon$-equilibrium strategy. Then the behavior cloning method cannot converge to the equilibrium strategy $\sigma^*$ based on $D'$ since BC cannot get the pure strategy for Player 1 under the influence of the data point $((I_1^{t_1} = I_1, I_2^{t_1} = \emptyset, \emptyset, 1, \{a_1, a_2\}), a_2, (I_1^{t_2} = I_1 a_2, I_2^{t_2} = I_2, \emptyset, 2, \{b_1, b_2\}), (0, 0), 0)$. Therefore, a slight violation of the equilibrium coverage assumption would cause failure in computing the $\epsilon$-equilibrium strategy of the original game using our behavior cloning method. In conclusion, the equilibrium coverage assumption is the minimal dataset coverage assumption that guarantees the convergence to the equilibrium strategy of the original game using our behavior cloning

method. Formally, $\sigma_{BC(\mathcal{D})}$ is guaranteed to be an $\epsilon$-equilibrium strategy of IIEFG $\mathcal{G}$ if and only if the offline dataset $\mathcal{D}$ is an $\epsilon$-equilibrium coverage of the IIEFG $\mathcal{G}$. $\qquad\square$

**Theorem D.10** (Theorem 4.6). *Let $\sigma_{BOMB(\mathcal{D})}$ represent the strategy profile learned by **BOMB** algorithm based on the offline dataset $\mathcal{D}$ with sufficient data under Assumption 4.3, $\sigma_{\mathcal{D}}$ represent the underlying behavior strategy of $\mathcal{D}$ and $\sigma^*$ represent the equilibrium strategy of IIEFG $\mathcal{G}$. Then the gap between $\sigma_{BOMB(\mathcal{D})}$ and $\sigma^*$ is guaranteed to be at most equal to, or smaller than, the gap between $\sigma_{\mathcal{D}}$ and $\sigma^*$. Formally, $\mathrm{GAP}(\sigma_{BOMB(\mathcal{D})}, \sigma^*) \leq \mathrm{GAP}(\sigma_{\mathcal{D}}, \sigma^*)$.*

*Proof.* According to Assumption 4.3, the error in training the behavior cloning strategy $\sigma_{BC(\mathcal{D})}$ from the dataset $\mathcal{D}$ is negligible. Therefore, by the behavior cloning process, $\sigma_{BC(\mathcal{D})}$ is identical to the behavior strategy underlying $\mathcal{D}$, i.e., $\sigma_{BC(\mathcal{D})} = \sigma_{\mathcal{D}}$. Then $\mathrm{GAP}(\sigma_{BOMB(\mathcal{D})}, \sigma^*) = \mathrm{GAP}(\sigma_{\mathcal{D}}, \sigma^*)$ if $\alpha = 1$ in our **BOMB algorithm**. If the dataset satisfies the uniform coverage, by Theorem 4.4, $\mathrm{GAP}(\sigma_{BOMB(\mathcal{D})}, \sigma^*) \leq \mathrm{GAP}(\sigma_{\mathcal{D}}, \sigma^*)$ if $\alpha = 0$ in our **BOMB algorithm**. Therefore, in general case, $\mathrm{GAP}(\sigma_{BOMB(\mathcal{D})}, \sigma^*) \leq \mathrm{GAP}(\sigma_{\mathcal{D}}, \sigma^*)$. $\qquad\square$

### D.2 GENERALIZATION BOUND FOR TRAINING MODEL

As described in the main paper, to conduct the BOMB framework, we need to train one behavior cloning policy and an environment model, which are both neural network models. Furthermore, these two models are trained in a supervised learning manner with different loss functions based on the offline EF dataset. Here, we provide a generalization bound for training such neural network models, facilitating the following analysis of the BOMB framework.

As we know, the supervised learning framework includes a data-generation distribution $\sigma$, a hypothesis class $\mathcal{H}$ of the neural network approximator, a training dataset $\mathcal{D}$, and evaluation metrics to evaluate the performance of any approximator. Here, we can use the loss function $l$ to evaluate the performance of any approximation. The learning framework aims to minimize the true risk function $L_\sigma(h)$ which is the expected loss function of $h \in \mathcal{H}$ under the distribution $\sigma$,

$$L_\sigma(h) = \mathbb{E}_{d \sim \sigma}[l(h(d), d)].$$

Accordingly, the empirical risk function $L_\mathcal{D}(h)$ on the training dataset $\mathcal{D}$ can be defined as:

$$L_\mathcal{D} = \frac{1}{|\mathcal{D}|} \sum_{d \sim \mathcal{D}} [l(h(d), d)].$$

To get a generalization bound, we use an auxiliary lemma from (Shalev-Shwartz & Ben-David, 2014). Therefore, we can measure the capacity of the composition function class $l \circ \mathcal{H}$ using the empirical Rademacher complexity on the training set $\mathcal{D}$ with size $m$, which is defined as:

$$\mathcal{R}_\mathcal{D}(l \circ \mathcal{H}) = \frac{1}{m} \mathbb{E}_{\mathbf{x} \sim \{+1, -1\}^m} [\sup_{h \in \mathcal{H}} \sum_{i=1}^{m} x_i \cdot l(h(d_i), d_i)]$$

where $\mathbf{x}$ is distributed i.i.d. according to uniform distribution in $\{+1, -1\}$. Before providing the generalization bound, we first provide the distance between two different approximators and a common theorem to facilitate the proof of the generalization bound.

**Definition D.11.** ($r$-cover) We say function class $\mathcal{H}_r$ $r$-cover $\mathcal{H}$ under $\ell_{\infty,1}$-distance if $\forall h, h \in \mathcal{H}$, there exists $h_r$ in $\mathcal{H}_r$ such that $||h - h_r||_{\infty,1} = \max_{x \in \mathcal{D}} ||h(x) - h_r(x)||_1 \leq r$.

**Definition D.12.** ($r$-covering number) The $r$-covering number of $\mathcal{H}$, $\mathcal{N}_{\infty,1}(\mathcal{H}, r)$, is the cardinality of the smallest function class $H_r$ that $r$-covers $\mathcal{H}$ under $\ell_{\infty,1}$-distance.

**Theorem D.13.** *(Shalev-Shwartz & Ben-David, 2014) Let $\mathcal{D}$ be a training set of size $m$ drawn i.i.d. from distribution $\sigma$. Then with probability of at least $1 - \delta$ over draw of $\mathcal{D}$ from $\sigma$, for all $h \in \mathcal{H}$,*

$$L_\sigma(h) - L_\mathcal{D}(h) \leq 2\mathcal{R}_\mathcal{D}(l \circ \mathcal{H}) + 4\sqrt{\frac{2 \ln(4/\delta)}{m}}.$$

We provide a bound to measure the generalizability of the trained approximator in a training dataset of size $m$ using the following theorem.

**Theorem D.14** (Generalization bound). *Assume that the loss function $l$ is $T$-Lipschitz continuous, then for hypothesis class $\mathcal{H}$ of approximator and distribution $\sigma$, with probability at least $1 - \delta$ over draw of the training set $\mathcal{D}$ with size $m$ from $\sigma$, for all $h \in \mathcal{H}$, we have*

$$L_\sigma(h) - L_\mathcal{D}(h) \leq 2 \cdot \inf_{r>0}\Big[\frac{\sqrt{2\log \mathcal{N}_{\infty,1}(\mathcal{H},r)}}{m} + Tr\Big] + 4\sqrt{\frac{2\ln(4/\delta)}{m}}.$$

*Proof.* According to Theorem D.13, we have

$$L_\sigma(h) - L_\mathcal{D}(h) \leq 2\mathcal{R}_\mathcal{D}(l \circ \mathcal{H}) + 4\sqrt{\frac{2\ln(4/\delta)}{m}}.$$

According to the assumption, the loss function $l(x, y)$ is $T$-Lipschitz continuous under $\ell_k$-distance, i.e., $|l(x, y) - l(x', y)| \leq T||x - x'||_k$, where $|| \cdot ||_k$ is the $k$-norm. Let $\mathcal{H}_r$ be the function class that $r$-cover $\mathcal{H}$ for some $r > 0$ and $|\mathcal{H}_r| = \mathcal{N}_{\infty,1}(\mathcal{H}, r)$ be the $r$-covering number of $\mathcal{H}_r$. For all $h \in \mathcal{H}$, $h_r \in \mathcal{H}_r$ is denoted to be the function approximator that $r$-covers $h$. Based on above equations, we have

$$|l(h(x), y) - l(h_r(x), y)| \leq T||h(x) - h_r(x)||_k \leq Tr.$$

Then we have

$$\mathcal{R}_\mathcal{D}(l \circ \mathcal{H}) = \frac{1}{m}\mathbb{E}_{\mathbf{x} \sim \{+1,-1\}^m}\Big[\sup_{h \in \mathcal{H}}\sum_{i=1}^{m} x_i \cdot l(h(d_i), d_i)\Big] \tag{1}$$

$$= \frac{1}{m}\mathbb{E}_{\mathbf{x} \sim \{+1,-1\}^m}\Big[\sup_{h \in \mathcal{H}}\sum_{i=1}^{m} x_i \cdot (l(h_r(d_i), d_i) + l(h(d_i), d_i) - l(h_r(d_i), d_i))\Big] \tag{2}$$

$$\leq \frac{1}{m}\mathbb{E}_{\mathbf{x} \sim \{+1,-1\}^m}\Big[\sup_{h_r \in \mathcal{H}_r}\sum_{i=1}^{m} x_i \cdot l(h_r(d_i), d_i)\Big] + \frac{1}{m}\mathbb{E}_{\mathbf{x} \sim \{+1,-1\}^m}\Big[\sup_{h \in \mathcal{H}}\sum_{i=1}^{m} |x_i \cdot Tr|\Big] \tag{3}$$

$$\leq \sup_{h_r \in \mathcal{H}_r}\sqrt{\sum_{i=1}^{m}(\ell(h_r, d_i))^2} \cdot \frac{\sqrt{2\log \mathcal{N}_{\infty,1}(\mathcal{H},r)}}{m} + \frac{Tr}{m}\mathbb{E}_\mathbf{x}||\mathbf{x}||_1 \tag{4}$$

$$\leq \frac{\sqrt{2\log \mathcal{N}_{\infty,1}(\mathcal{H},r)}}{m} + Tr \tag{5}$$

The reduction from Eq. 3 to Eq. 4 is based on Massart's lemma (Shalev-Shwartz & Ben-David, 2014). Finally,

$$L_\sigma(h) - L_\mathbf{D}(h) \leq 2\mathcal{R}_\mathcal{D}(l \circ \mathcal{H}) + 4\sqrt{\frac{2\ln(4/\delta)}{m}} \tag{6}$$

$$\leq 2 \cdot \inf_{r>0}\Big[\frac{\sqrt{2\log \mathcal{N}_{\infty,1}(\mathcal{H},r)}}{m} + Tr\Big] + 4\sqrt{\frac{2\ln(4/\delta)}{m}}$$

$\square$

Therefore, given a training dataset of size $m$, we have a generalization bound for the error depending on the characteristics of the loss function. In this paper, we follow the supervised learning framework to train the behavior cloning policy and environment model. Therefore, we can provide the following assumptions for the trained policy and environment models based on the above theorem.

**Assumption D.15.** Suppose the error for training the behavior cloning policy is less than an extremely small $\epsilon$ on the dataset with enough data (the size of data can be computed according to the above theorem). In that case, we consider that the trained behavior cloning policy is the same as the underlying behavior strategy of the dataset.

**Assumption D.16.** Suppose the error for training the environment model is less than an extremely small $\epsilon$ on the dataset with enough data. In that case, we consider that the trained environment model can provide the full information for the underlying game of the dataset.

# E   IMPLEMENTATION DETAILS

Here, we provide the details for the model-based method by introducing our instantiable algorithms: MB-PSRO and MB-CFR, which are adaptations from two widely-used online equilibrium finding algorithms, PSRO and Deep CFR.

## E.1   MB-PSRO

---

**Algorithm 2** MB-PSRO

---

1: **Input:** Trained environment model $E_{\theta_e}$
2: Initial policy sets $\Pi$ for all players;
3: Compute expected rewards $U^\Pi$ for each strategy $\pi \in \Pi$ *based on the environment model $E_{\theta_e}$*;
4: Initialize mate-strategies $\sigma_i = \text{UNIFORM}(\Pi_i)$, $\forall i$;
5: **repeat**
6:   **for** each player $i \in [1, .., n]$ **do**
7:     **for** best response episodes $t \in [1, ..., T]$ **do**
8:       Sample $\pi_{-i} \sim \sigma_{-i}$;
9:       Train best response policy $\pi_i'$ over $\rho \sim (\pi_i', \pi_{-i})$, which *samples on the environment model $E_{\theta_e}$*;
10:    **end for**
11:    add the best response policy $\pi_i'$ to policy set $\Pi_i$;
12:  **end for**
13:  Compute missing entries in $U^\Pi$ *based on the environment model $E_{\theta_e}$*;
14:  Compute the meta-strategy $\sigma$ using any meta-solver;
15: **until** Meet the convergence condition
16: **Output:** Policy set $\Pi$ and meta-strategy $\sigma$

---

We present the whole framework in Alg. 2. In the beginning, we need a well-trained environment model $E_{\theta_e}$ as input to replace the function of the actual environment. Firstly, we initialize policy sets $\Pi$ for all players using random strategies. Then, we estimate the expected utilities for each strategy profile based on the model $E_{\theta_e}$ to form the meta-game matrix. In vanilla PSRO, this process needs to interact with the actual game environment. However, in the offline setting, the actual game environment is not available. Therefore, we use the well-trained environment model $E_{\theta_e}$ to replace the actual game environment to provide the information needed in the algorithm. After building the meta-game matrix, the meta-strategy is initialized by a uniform strategy. Next, we compute the best response policy for every player and add these trained best response policies to their policy sets. When training the best response policy oracle using DQN or other RL algorithms, we sample the training data based on the environment model $E_{\theta_e}$. After adding trained best response policies, we compute missing entries in the meta-game matrix based on the trained environment model $E_{\theta_e}$. Then, the meta-strategy $\sigma$ of the meta-game matrix can be computed using any meta-solver, such as the Nash solver or $\alpha$-rank algorithm. For games with more than two players, the $\alpha$-rank algorithm is taken as the meta-solver. Finally, we repeat the above processes until meeting the convergence condition and output the policy set and meta-strategy as the approximate equilibrium strategy.

To compute the CCE strategy, we also instantiate one algorithm: MB-JPSRO, an adaptation from the JPSRO algorithm. The process of JPSRO is similar to PSRO except for the best response computation and meta solver. Therefore, MB-JPSRO is also similar to MB-PSRO. For this reason, we do not cover MB-JPSRO in detail here.

## E.2   MB-CFR

Alg. 3 shows the process of MB-CFR, which is adapted from the Deep CFR algorithm. It also needs the well-trained environment model $E_{\theta_e}$ as input for the MB-CFR algorithm. We first initialize regret and strategy networks for each player and then initialize regret and strategy memories for each player (Lines 2-4). Then we need to update the regret network for every player. To do this, we perform a traverse function to collect corresponding training data. The traverse function can be any sampling-based CFR algorithm. Here, we use the external sampling algorithm as the traverse method to collect training data, and the process of external sampling is shown in Alg. 4. In this

**Algorithm 3** MB-CFR

1: **Input:** Trained environment model $E_{\theta_e}$
2: Initialize regret network $R(I, a|\theta_{r,p})$ for all players;
3: Initialize strategy network $S(I|\theta_{\pi,p})$ for all players;
4: Initialize regret memory $M_{r,p}$ and strategy memory $M_{\pi,p}$ for every player $p$;
5: **for** iteration $t = 1$ to $T$ **do**
6:   **for** player $p \in [1, ..., n]$ **do**
7:    **for** traverse episodes $k \in [1, ..., K]$ **do**
8:     **TRVERSE**$(\phi, p, \theta_{r,p}, \theta_{\pi,-p}, M_{r,p}, M_{\pi,-p}, E_{\theta_e})$;
    *# Use sample algorithm to traverse game tree, record regret and strategy training data*
9:    **end for**
10:    Train $\theta_{r,p}$ from scratch based on regret memory $M_{r,p}$ for every player $p$;
11:   **end for**
12: **end for**
13: Train $\theta_{\pi,p}$ based on strategy memory $M_{\pi,p}$ for every player $p$;
14: **Output:**$\theta_{\pi,p}$ for every player $p$

---

**Algorithm 4** TRVERSE$(s, p, \theta_{r,p}, \theta_{\pi,-p}, M_{r,p}, M_{\pi,-p}, E_{\theta_e})$-External Sampling Algorithm

1: **if** $s$ is terminal state **then**
2:   Get the utility $u_p(s)$ from the environment model $E_{\theta_e}$;
3:   **Output:** $u_p(s)$
4: **else if** $s$ is a chance state **then**
5:   Sample an action $a$ from the available actions, which is obtained from model $E_{\theta_e}$;
6:   $s' = E_{\theta_e}(s, a)$;
7:   **Output:**TRAVERSE$(s', p, \theta_{r,p}, \theta_{\pi,-p}, M_{r,p}, M_{\pi,-p}, E_{\theta_e})$
8: **else if** $P(s) = p$ **then**
9:   $I \leftarrow s[p]$; *# Get the corresponding information set from the game state*
10:   $\sigma(I) \leftarrow$ strategy of $I$ computed using regret values $R(I, a|\theta_{r,p})$ based on regret matching;
11:   **for** $a \in A(s)$ **do**
12:    $s' = E_{\theta_e}(s, a)$;
13:    $u(a) \leftarrow$ TRAVERSE$(s', p, \theta_{r,p}, \theta_{\pi,-p}, M_{r,p}, M_{\pi,-p}, E_{\theta_e})$;
14:   **end for**
15:   $u_\sigma \leftarrow \sum_{a \in A(s)} \sigma(I, a)u(a)$;
16:   **for** $a \in A(s)$ **do**
17:    $r(I, a) \leftarrow u(a) - u_\sigma$;
18:   **end for**
19:   Insert the infoset and its action regret values $(I, r(I))$ into regret memory $M_{r,p}$;
20:   **Output:** $u_\sigma$
21: **else**
22:   $I \leftarrow s[p]$;
23:   $\sigma(s) \leftarrow$ strategy of $I$ computed using regret value $R(I, a|\theta_{r,-p})$ based on regret matching;
24:   Insert the infoset and its strategy $(I, \sigma(s))$ into strategy memory $M_{\pi,-p}$;
25:   Sample an action $a$ from distribution $\sigma(s)$;
26:   $s' = E_{\theta_e}(s, a)$;
27:   **Output:** TRAVERSE$(s', p, \theta_{r,p}, \theta_{\pi,-p}, M_{r,p}, M_{\pi,-p}, E_{\theta_e})$;
28: **end if**

---

traverse function, we collect the regret training data of the traveler, and the strategy training data of other players is also gathered. After performing the traverse function several times, the regret network can be updated based on the regret memory using a supervised learning algorithm. The above processes are repeated for $T$ times. Then, the average strategy network for every player is trained based on its corresponding strategy memory. Finally, the trained average strategy networks are output as the approximate equilibrium strategy.

# F    ADDITIONAL EXPERIMENTAL RESULTS

In this section, we provide more experimental results and an ablation study. Finally, we provide the main parameters we used in our experiments.

## F.1    EXPERIMENTAL RESULTS

Here, we first verify that the performance of the model-based approach is independent of the algorithm used for computing equilibrium strategy. To this end, we perform both MB-CFR and MB-PSRO algorithms in the two-player Kuhn poker game under different sizes of offline datasets. Fig. 19 shows the results. We can find that under the same size of an offline dataset, MB-PSRO and MB-CFR achieve nearly identical results. When the size of the offline dataset increases, the performance of both algorithms becomes better. It may be caused by the environment model being well-trained with more data.

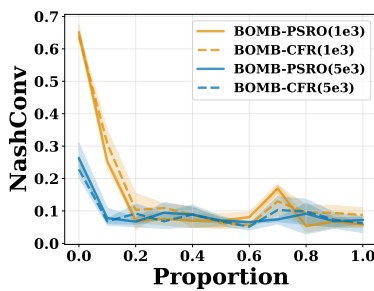

Figure 19: Results of different MB methods

These observations indicate that the performance of the model-based algorithm is independent of the algorithm used to compute the equilibrium strategy and mainly relies on the similarity between the trained environment model and the actual environment.

Then, we provide experimental results for the BOMB framework on multi-player games, including four-player and five-player Kuhn poker games, to compute NE strategies. The results, shown in Fig. 20, are consistent with those in the main paper. Across all cases, our BOMB framework outperforms or matches the performance of both BC and MB algorithms, aligning with the theoretical analysis. As the proportion of the random dataset increases, the performance of BC decreases, while the performance of MB is unstable and shows a slight downward trend. Here, we use OEF-CFR as the model-based algorithm, noting that CFR-based algorithms do not guarantee convergence to NE strategies in multi-player games. Additionally, the performance of the MB framework relies on the quality of the trained environment model. Poor performance in the MB algorithm may stem from either an inadequately trained environment model or the limitations of CFR-based algorithms in multi-player settings. This underscores the challenges of learning a robust strategy and training an accurate environment model in these complex games. Fig. 20 also illustrates the weights of BC in the BOMB algorithm under different datasets. In the four-player Kuhn poker game, the BC weight quickly drops to zero as the proportion of the random dataset increases, indicating that MB outperforms BC in most scenarios except when the random dataset proportion is low. Conversely, in the five-player Kuhn poker game, the BC weight remains high in most cases unless the random dataset proportion is very high. This is because the MB framework performs poorly in these highly dynamic multi-player games, making it a significant challenge to approximate equilibrium strategies with the MB method. These findings highlight the importance of improving both the strategy-learning process and the environment model in multi-player settings.

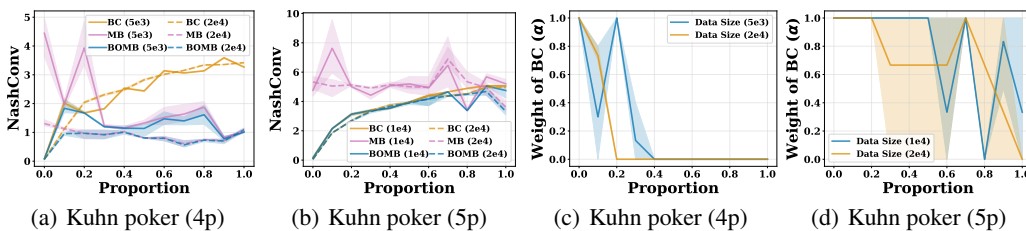

(a) Kuhn poker (4p)     (b) Kuhn poker (5p)     (c) Kuhn poker (4p)     (d) Kuhn poker (5p)

Figure 20: Performance of BOMB on multi-player games.

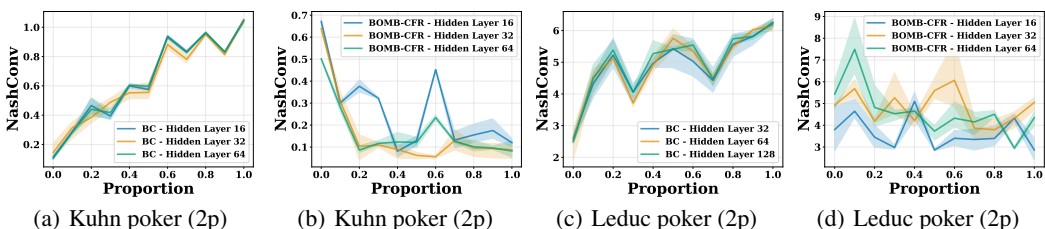

Figure 21: Abalation results for different hidden layer sizes.

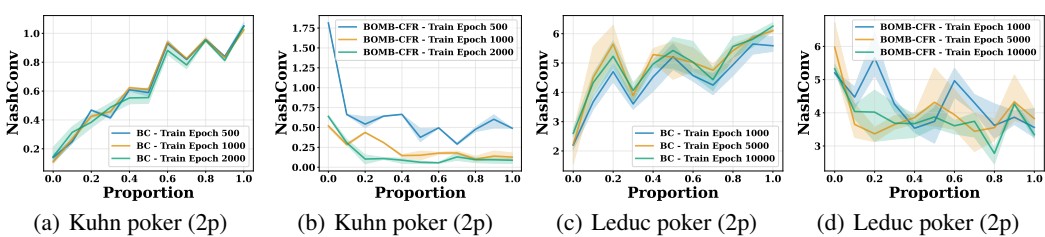

Figure 22: Abalation results for different train epochs.

## F.2 ABLATION STUDY

To investigate the influence of hyperparameters, we conduct several ablation experiments on two-player Kuhn poker and Leduc poker games. We consider different model structures with various numbers of hidden layers. Specifically, for the 2-Player Kuhn poker game, we use different environment models with 16, 32, and 64 hidden layers. For the 2-Player Leduc poker game, which is a more complicated game, the numbers of hidden layers for different models are 32, 64, and 128. In addition, we train the environment models for different epochs to evaluate the robustness of our approach. Figs. 21-22 show these ablation results. We find that the number of hidden layers and the number of training epochs have little effect on the performance of the BC algorithm. These results further verify that the performance of the BC algorithm primarily depends on the quality of the dataset. As we know, the performance of the model-based method mainly depends on the trained environment model. Since the number of the hidden layer and the number of training epochs influence the training phase of the environment model, the number of the hidden layer and the number of train epochs have a slight impact on the performance of the model-based method. As long as the size of the hidden layer and the number of training epochs can guarantee that the environment model is trained accurately, the performance of the model-based method will not be affected.

## F.3 PARAMETER SETTING

We list the parameters used to train the behavior cloning policy and environment model for all games used in our experiments in Tab. 2.

| Methods | Behavior Cloning Algorithm | | | | Environment Model Training | | | |
|---|---|---|---|---|---|---|---|---|
| Games | Kuhn Poker (2p) | | Kuhn Poker (3p) | | Kuhn Poker (2p) | | Kuhn Poker (3p) | |
| Data size | 500 | 1000 | 1000 | 5000 | 500 | 1000 | 1000 | 5000 |
| Hidden layer | 32 | 32 | 32 | 32 | 32 | 32 | 32 | 32 |
| Batch size | 32 | 32 | 32 | 32 | 32 | 32 | 32 | 32 |
| Train epoch | 1000 | 2000 | 5000 | 5000 | 1000 | 2000 | 2000 | 5000 |
| Games | Kuhn Poker (4p) | | Kuhn Poker (5p) | | Kuhn Poker (4p) | | Kuhn Poker (5p) | |
| Data size | 5000 | 20000 | 10000 | 20000 | 5000 | 20000 | 10000 | 20000 |
| Hidden layer | 64 | 64 | 64 | 64 | 64 | 64 | 64 | 64 |
| Batch size | 64 | 128 | 128 | 128 | 64 | 128 | 128 | 128 |
| Train epoch | 5000 | 5000 | 5000 | 5000 | 5000 | 5000 | 5000 | 5000 |
| Games | Leduc Poker (2p) | | Leduc Poker (3p) | | Leduc Poker (2p) | | Leduc Poker (3p) | |
| Data size | 10000 | 20000 | 10000 | 20000 | 10000 | 20000 | 10000 | 20000 |
| Hidden layer | 128 | 128 | 128 | 128 | 64 | 128 | 128 | 128 |
| Batch size | 128 | 128 | 128 | 128 | 64 | 128 | 128 | 128 |
| Train epoch | 10000 | 10000 | 10000 | 5000 | 10000 | 10000 | 10000 | 10000 |
| Games | Liar's Dice | | Phantom TTT | | Liar's Dice | | Phantom TTT | |
| Data size | 10000 | 20000 | 10000 | 20000 | 10000 | 20000 | 10000 | 20000 |
| Hidden layer | 64 | 64 | 128 | 128 | 64 | 64 | 128 | 128 |
| Batch size | 128 | 128 | 128 | 128 | 64 | 128 | 128 | 128 |
| Train epoch | 5000 | 5000 | 5000 | 5000 | 5000 | 5000 | 5000 | 5000 |

Table 2: Parameters for Behavior Cloning algorithm and Environment Model Training

