# OpenReview forum: "Offline Equilibrium Finding in Extensive-form Games: Datasets, Methods, and Analysis"
_ICLR.cc/2026/Conference — Submitted to ICLR 2026_

### Official Review · Reviewer_XxFT · 2025-10-17

**Soundness:** 2
**Presentation:** 2
**Contribution:** 1
**Rating:** 2
**Confidence:** 5

**Summary:**

This paper introduces the problem of Offline Equilibrium Finding (Offline EF) in imperfect-information extensive-form games (IIEFGs), where the goal is to compute equilibrium strategies using only pre-collected datasets without environment interaction. The authors propose the BOMB framework, which combines behavior cloning (BC) and model-based (MB) methods, and provide theoretical guarantees under strong dataset coverage assumptions. They also contribute a set of offline datasets for benchmarking, collected using random, expert, and learning-based policies. Extensive experiments are conducted on small-scale games to validate BOMB against offline RL baselines and its individual components.

**Strengths:**

1. Offline EF is a meaningful and underexplored direction that bridges offline learning and game theory. The paper clearly motivates the problem and differentiates it from offline RL and online equilibrium finding.

2. The paper includes extensive experiments covering two-player, multi-player, and hybrid dataset settings.

**Weaknesses:**

1. The theoretical guarantees rely on strong assumptions such as uniform coverage or ε-equilibrium coverage, which are impractical in real-world settings. The analysis does not address more realistic, partial-coverage settings. In Theorem 4.2, dataset $\mathcal D$ requires sufficient state reached, and the BC strategy is directly weighted into dataset $\mathcal D$. This implies that the $\varepsilon$-equilibrium is bounded with $\alpha$, and if $\alpha$ is too small, $|\mathcal D|$ must be very large. Moreover, I find the proof section of the main text to be rather redundant.

2. All experiments are conducted on small-scale games that can be solved exactly with tabular methods. There is no evaluation on larger games or real human gameplay data, limiting the claim of practical impact.

3. The datasets are generated using high-variance RL algorithms (e.g., Deep CFR, PSRO), which contradicts the motivation of avoiding simulators and may not reflect real-world data distributions.

4. The NashConv values are high compared to standard online methods, and the performance of BOMB in fully offline settings without fine-tune remains unclear.

5. Overstated claim in (Line 52) " there has been no dedicated study addressing the offline setting in multi-player games". See [1-12]. I haven't read all the papers in recent years, so there should be more similar works.

6. The algorithm BOMB is not novel in IIEFGs. Combine behavior strategy and equilibrium strategy is a common refinement in solving IIEFGs [13, 14]. The algorithm appears rather inefficient, and the training overhead seems quite substantial.

7. The BC strategy and MB strategy appear to be trained separately, which leads to a problem: the MB strategy cannot handle the slight suboptimality of the EFG structure. For instance, P1 optimal strategy should be to execute action A (with an expected value of +1), but if P1 execute action B (with an expected value of +0.99), the MB strategy will not know how to respond when Player 1 executes action B. For instance, if P1 acts B, then Player 2's expected value for action C is -1 and action D is 0, Player 2's MB strategy would clearly choose action D. However, within the BOMB framework, this MB strategy would not be sampled at all.

8. Relevant works have been compiled only up to 2021. In recent years, considerable work has been undertaken towards resolving IIEFG, including [15-27].

[1] Yuheng Jing, Kai Li, Bingyun Liu, Yifan Zang, Haobo Fu, Qiang Fu, Junliang Xing, Jian Cheng. Towards Offline Opponent Modeling with In-context Learning. ICLR 2024

[2] Hang Xu, Kai Li, Haobo Fu, Qiang Fu, Junliang Xing, Jian Cheng. Dynamic Discounted Counterfactual Regret Minimization. ICLR 2024

[3] Jingxiao Chen, Weiji Xie, Weinan Zhang, Yong Yu, Ying Wen. Offline Fictitious Self-Play for Competitive Games. arxiv 2024

[4] Runyu Lu, Yuanheng Zhu, Dongbin Zhao. Constrained Exploitability Descent: An Offline Reinforcement Learning Method for Finding Mixed-Strategy Nash Equilibrium. ICML 2025

[5] Junren Luo, Wanpeng Zhang, Mingwo Zou, Jiongming Su, Jing Chen. Offline PSRO with Max-Min Entropy for Multi-Player Imperfect Information Games. 2022 China Automation Congress

[6] Hai Zhong, Xun Wang, Zhuoran Li, Longbo Huang. Offline-to-Online Multi-Agent Reinforcement Learning with Offline Value Function Memory and Sequential Exploration. AAMAS 2025

[7] Weijun Zeng, Yinghao Li, Xiaosi Chen, Zijie Chang, Fei Ge. GNN-ReBeL: Enhancing Neural Belief Representation for Imperfect-Information Games. IEEE SMC 2025

[8] Weijun Zeng, Yinghao Li, Xiaosi Chen, Zijie Chang, Fei Ge. An Investigation of Subgame Depth in ReBeL: Impact on Convergence and Performance in Imperfect-Information Games. IEEE SMC 2025

[9] David Sychrovský, Michal Šustr, Elnaz Davoodi, Michael Bowling, Marc Lanctot, Martin Schmid. Learning Not to Regret. AAAI 2024

[10] David Sychrovský, Martin Schmid, Michal Šustr, Michael Bowling. Meta-Learning in Self-Play Regret Minimization. arxiv 2025

[11] Fuxiang Zhang, Chengxing Jia, Yi-Chen Li, Lei Yuan, Yang Yu, Zongzhang Zhang. Discovering Generalizable Multi-agent Coordination Skills from Multi-task Offline Data. ICLR 2023

[12] Zhengdao Shao, Liansheng Zhuang, Houqiang Li, Shafei Wang. COPSRO: An Offline Empirical Game Theoretic Method With Conservative Critic. IEEE TNNLS 2025

[13] Gabriele Farina, Christian Kroer, Tuomas Sandholm. Regret Minimization in Behaviorally-Constrained Zero-Sum Games. ICML 2017

[14] Brian Hu Zhang, Tuomas Sandholm. On the Outcome Equivalence of Extensive-Form and Behavioral Correlated Equilibria. AAAI 2024

[15] Noam Brown, Anton Bakhtin, Adam Lerer, Qucheng Gong. Combining Deep Reinforcement Learning and Search for Imperfect-Information Games. NeurIPS 2020

[16] Gabriele Farina, Christian Kroer, Tuomas Sandholm. Faster Game Solving via Predictive Blackwell Approachability: Connecting Regret Matching and Mirror Descent. AAAI 2021

[17] Brian Hu Zhang, Tuomas Sandholm. Subgame solving without common knowledge. NeurIPS 2021

[18] Hang Xu, Kai Li, Haobo Fu, Qiang Fu, Junliang Xing. AutoCFR: Learning to Design Counterfactual Regret Minimization Algorithms. AAAI 2022

[19] Enmin Zhao, Renye Yan, Jinqiu Li, Kai Li, Junliang Xing. AlphaHoldem: High-Performance Artificial Intelligence for Heads-Up No-Limit Poker via End-to-End Reinforcement Learning. AAAI 2022

[20] Martin Schmid, Kevin Waugh, Matej Moravčík, Nolan Bard, Neil Burch, Rudolf Kadlec, Finbarr Timbers, Marc Lanctot, Josh Davidson, G. Zacharias Holland, Elnaz Davoodi, Alden Christianson, Michael Bowling. Student of games: A unified learning algorithm for both perfect and imperfect information games. Science 2023

[21] Stephen Marcus McAleer, Gabriele Farina, Marc Lanctot, Tuomas Sandholm. ESCHER: Eschewing Importance Sampling in Games by Computing a History Value Function to Estimate Regret. ICLR 2023

[22] Weiming Liu, Haobo Fu, Qiang Fu, Wei Yang. Opponent-Limited Online Search for Imperfect Information Games. ICML 2023

[23] Linjian Meng, Zhenxing Ge, Pinzhuo Tian, Bo An, Yang Gao. An Efficient Deep Reinforcement Learning Algorithm for Solving Imperfect Information Extensive-Form Games. AAAI 2023

[24] Boning Li, Zhixuan Fang, Longbo Huang. RL-CFR: Improving Action Abstraction for Imperfect Information Extensive-Form Games with Reinforcement Learning. ICML 2024

[25] Yuheng Jing, Bingyun Liu, Kai Li, Yifan Zang, Haobo Fu, Qiang Fu, Junliang Xing, Jian Cheng. Opponent Modeling with In-context Search. NeurIPS 2024

[26] Boning Li, Longbo Huang. Efficient Online Pruning and Abstraction for Imperfect Information Extensive-Form Games. ICLR 2025

[27] Linjian Meng, Tianpei Yang, Youzhi Zhang, Zhenxing Ge, Shangdong Yang, Tianyu Ding, Wenbin Li, Bo An, Yang Gao. Efficient Last-Iterate Convergence of Counterfactual Regret Minimization Algorithms. NeurIPS 2025

[28] Matthew Thomas Jackson, Uljad Berdica, Jarek Luca Liesen, Shimon Whiteson, Jakob Nicolaus Foerster. A Clean Slate for Offline Reinforcement Learning. NeuIPS 2025

**Questions:**

1. Definition 2.1 assumes a unique equilibrium strategy $\sigma^*$. How is the gap metric defined when multiple equilibria exist? I found this metric unreasonable. Why not use NashConv?

2. The state representation includes information sets, game-level info, and available actions. How does BOMB encoding different states in various games?

3. What are the computational and statistical barriers to applying BOMB to larger games, and how can they be addressed?

4. How does BOMB compare to recent online methods like Deep CFR and PSRO? Can we fine-tune BOMB model to achieve better performance?

5. Can you show results against MoBRAC [28]?

---

> ### Author Response · Authors · 2025-11-25
> **Response to Reviewer XxFT (1/4)**
>
> We appreciate the reviewer’s comments. Here we provide our response to the weaknesses and your questions.
>
> ----
>
> ***Q: The theoretical guarantees rely on strong assumptions such as uniform coverage or $\epsilon$-equilibrium coverage, which are impractical in real-world settings.***
>
> **Reply:** We would like to clarify several misunderstandings regarding our theoretical results.
> - Our theoretical analysis indeed relies on some dataset assumptions. However, these assumptions are introduced only as sufficient conditions that allow us to derive upper-bound guarantees. This follows the standard practice in offline RL, where concentrability or coverage assumptions are used for theoretical analysis. We do not claim that real datasets must satisfy uniform coverage or $\epsilon$-equilibrium coverage assumptions. These assumptions formalize when MB or BC can be provably near equilibrium in a fully offline setting.
> - Actually, we have explicitly studied partial coverage settings in experiments. We evaluate our method on random, learning, expert, and hybrid offline datasets, which correspond to various realistic partial coverage cases. The results also show how BC and MB behave under these partial coverage datasets.
> - We want to highlight that the \alpha coefficient is used only in the BOMB combination mechanism and does not appear in the theoretical guarantees of BC or MB. Therefore, it is incorrect to accuse that $\epsilon$-equilibrium is bounded with $\alpha$, and if $\alpha$ is too small, $|D|$ must be very large.
>
> ----
>
> ***Q: All experiments are conducted on small-scale games that can be solved exactly with tabular methods. There is no evaluation on larger games or real human gameplay data, limiting the claim of practical impact.***
>
> **Reply:**
> - Except for two-player Kuhn Poker and two-player Leduc Poker, which can be solved with tabular methods, we also include Liar’s Dice, Phantom Tic Tac Toe, and multiplayer variants, all of which introduce large information sets that make exact tabular solutions computationally infeasible. These environments are widely used as standard evaluation testbeds in imperfect information extensive form games precisely because they are nontrivial and exhibit strategic complexity.
> - The goal of offline equilibrium finding is to study how to compute equilibrium strategies when only an offline dataset is available. For this purpose, using environments with known equilibria is essential since it allows us to quantitatively measure approximation quality and equilibrium gap. Large-scale games with unknown equilibria would prevent reliable evaluation and comparison. Our chosen suite balances structural complexity with evaluability and follows established practice in the offline setting.
> - Our work focuses on the methodological and theoretical foundations of offline equilibrium finding. The proposed datasets and benchmarks already cover diverse partial coverage scenarios, multiple data generation mechanisms, and both two-player and multiplayer settings. These are sufficient for demonstrating the effectiveness and robustness of the proposed framework.
>
> ----
>
> ***Q: The datasets are generated using high-variance RL algorithms (e.g., Deep CFR, PSRO), which contradicts the motivation of avoiding simulators and may not reflect real-world data distributions.***
>
> **Reply:** We would like to clarify several misunderstandings about dataset generation.
> - We would like to emphasize that dataset generation is only the preparation phase for the offline equilibrium finding setting. The objective of offline equilibrium finding is to study how to compute equilibria when the dataset is fixed, and no further simulator interaction is available. This objective does not impose constraints on how the dataset was originally produced, similar to established offline RL benchmarks, where expert demonstrations or trajectories generated by prior algorithms are standard. Incorporating expert policies for part of the dataset follows common offline learning research and remains fully consistent with the offline setting. Employing multiple data generation mechanisms is essential for studying the offline equilibrium problem.
> - Employing multiple data generation mechanisms is essential for studying the offline equilibrium finding problem. In this work, we generate three distinct types of datasets: random dataset, learning dataset, and expert dataset. Each type corresponds to different real-world scenarios as discussed in the paper. The dataset, therefore, captures a broad set of realistic cases. This design allows our datasets to model diverse datasets encountered in real applications.

---

> ### Author Response · Authors · 2025-11-25
> **Response to Reviewer XxFT (2/4)**
>
> ***Q: The NashConv values are high compared to standard online methods, and the performance of BOMB in fully offline settings without fine-tune remains unclear.***
>
> **Reply:**
> - Comparing with online equilibrium finding methods is not appropriate. Online equilibrium finding algorithms such as CFR and PSRO can access the game simulator and update their strategy exactly. In contrast, our setting is fully offline, where algorithms operate solely on an offline dataset with even partial data coverage. NashConv values in the offline setting are therefore inherently larger than those in the online setting.
> - BOMB with the random combination method is fully offline, i.e., no simulator interaction and no online fine-tuning are used. Figure 6 shows the results.
>
> ----
>
> ***Q: Overstated claim in (Line 52) " there has been no dedicated study addressing the offline setting in multi-player games". See [1-12]. I haven't read all the papers in recent years, so there should be more similar works.***
>
> **Reply:** We appreciate the reviewer’s comments. The original phrasing may appear too strong when interpreted broadly. We will revise the sentence to provide a more precise description of prior work. Here, we want to clarify the distinction between our paper and these existing works.
> - Most of the cited works in [1-12] do not address the problem studied in our paper, i.e., offline equilibrium finding in imperfect-information extensive-form games. Many of these papers focus on offline multi-agent reinforcement learning, offline opponent modelling, or offline self-play. These settings differ fundamentally from our formulation. To our knowledge, none of these works provides a general framework for computing approximate Nash or correlated equilibria directly from only an offline dataset in imperfect-information extensive-form games.
> - Our paper introduces a comprehensive formulation and analysis of the offline equilibrium finding problem in imperfect-information extensive-form games. We provide a benchmark dataset, a practical offline equilibrium finding algorithm, and a comprehensive theoretical analysis for the offline equilibrium finding problem in extensive form games. This contribution remains distinct from the cited works and complements rather than overlaps with existing research.
>
> ----
>
> ***Q: The algorithm BOMB is not novel in IIEFGs. Combine behavior strategy and equilibrium strategy is a common refinement in solving IIEFGs [13, 14]. The algorithm appears rather inefficient, and the training overhead seems quite substantial.***
>
> **Reply:**
> - We want to clarify that the works cited in [13-14] address theoretical properties of behaviourally constrained equilibria or outcome equivalence between extensive form and behavioural correlated equilibria. These studies do not propose algorithms for offline equilibrium finding, do not operate on offline datasets, and do not consider the problem of computing equilibrium without simulator interaction. In contrast, our BOMB is a practical framework designed specifically for the offline equilibrium finding setting, where the algorithm can only rely on an offline dataset.
> - The combination referred to in the cited works is a theoretical refinement regarding the representation of equilibria, not a learnable combination mechanism between behaviour cloning and the model-based framework. BOMB introduces the combination of BC and MB with a coefficient \alpha and provides both a model-based framework and a behaviour cloning method that can be trained in a fully offline manner. This combination is not present in the cited works, nor in existing equilibrium algorithms.
>
> ----
>
> ***Q: The BC strategy and MB strategy appear to be trained separately, which leads to a problem:...***
>
> **Reply:** We would like to clarify several misunderstandings about the MB strategy.
> - In the MB method, we first train an environment model from the offline dataset. This model captures only the game dynamics, that is, the state transition function, and does not encode any behavioural bias from the dataset. We then apply MB CFR or MB PSRO on the trained environment model to obtain an approximate equilibrium strategy, which serves as the MB strategy. As discussed in Appendix D (Theorem D.5 and Theorem D.7), slight violations of the uniform coverage assumption may affect the theoretical guarantees. However, in practice, because the environment model is implemented as a neural network, it generalizes to unseen states and provides reasonable predictions for states not directly covered in the dataset.
> - In BOMB, BC and MB are trained separately by design because they capture complementary strengths, as demonstrated in our experimental results. Their combination through the coefficient \alpha ensures that the final policy benefits from both methods and leverages their respective advantages.

---

> ### Author Response · Authors · 2025-11-25
> **Response to Reviewer XxFT (3/4)**
>
> ***Q: Relevant works have been compiled only up to 2021. In recent years, considerable work has been undertaken towards resolving IIEFG, including [15-27].***
>
> **Reply:** We thank the reviewer for the pointers to recent work and would like to clarify the scope of our literature review.
> - The papers cited in [15 -27] indeed represent important advances in solving imperfect information extensive form games. However, these works focus on online equilibrium finding settings, all of which assume access to simulators or environment interaction during training.
> - Our paper studies the offline equilibrium finding problem in imperfect-information extensive-form games, where the algorithm must compute equilibria solely from an offline dataset with no further interaction with the environment. To the best of our knowledge, none of the works listed in [15-27] addresses this offline setting, nor do they investigate equilibrium finding under offline datasets with partial dataset coverage.
>
> ***Q: Definition 2.1 assumes a unique equilibrium strategy \sigma*. How is the gap metric defined when multiple equilibria exist? I found this metric unreasonable. Why not use NashConv?***
>
> **Reply:** The GAP(·) in Definition 2.1 is intended as a general placeholder for any equilibrium gap metric used in imperfect information extensive form games. Definition 2.1 does not assume a specific type of equilibrium and metric function. Rather, it formalizes the notion of evaluating how far a candidate strategy deviates from some target equilibrium concept. In Lines 136 to 138, we explicitly instantiate this general form with multiple standard metrics, including NashConv for NE and (C)CE Gap Sum for (C)CE. NashConv is indeed one of the primary metrics we use, and it is reported in all experiments where NE concepts apply. The abstract definition in Definition 2.1 is designed to keep the theory general and to accommodate different equilibrium notions across different settings.
>
> ----
>
> ***Q: The state representation includes information sets, game-level info, and available actions. How does BOMB encoding different states in various games?***
>
> **Reply:** Our BOMB algorithm is not designed to generalize across different games. The goal of BOMB is to compute an approximate equilibrium for a single imperfect information extensive form game given its fixed offline dataset. Therefore, for each game, we apply BOMB independently using that game’s specific state representation.
>
> ----
>
> ***Q: What are the computational and statistical barriers to applying BOMB to larger games, and how can they be addressed?***
>
> **Reply:**
> - When scaling BOMB to larger games, both computational and statistical challenges arise. For the BC technique, larger games typically require substantially more offline data to accurately cover the large information sets and to train a reliable policy.
> - For the MB method, the primary computational cost stems from training the environment model and then running MB-CFR or MB-PSRO on the trained environment model. As the game size grows, the number of information sets, actions, and transition functions increases, which significantly raises the complexity of both model learning and equilibrium computation.
> - Addressing these limitations is an important direction for future work. Possible approaches include more efficient model architectures and scalable equilibrium-finding algorithms tailored to the offline setting.
>
> ----
>
> ***Q: How does BOMB compare to recent online methods like Deep CFR and PSRO? Can we fine-tune BOMB model to achieve better performance?***
>
> **Reply:**
> - BOMB and recent online methods, such as Deep CFR and PSRO, address fundamentally different problem settings. Deep CFR and PSRO operate in the online setting and require repeated interaction with a simulator to update strategies. In contrast, BOMB is designed specifically for the offline equilibrium finding setting, where the algorithm must operate strictly on an offline dataset without any simulator interaction. For this reason, it is not meaningful to compare offline equilibrium finding methods directly with online approaches that rely on full environment access.
> - Regarding fine-tuning, BOMB is not a model that can be fine-tuned in the conventional sense, because its final strategy is a mixture of the MB and BC strategies rather than a single parameterized network that can be further optimized through gradient-based updates.

---

> ### Author Response · Authors · 2025-11-25
> **Response to Reviewer XxFT (4/4)**
>
> ***Q: Can you show results against MoBRAC [28]?***
>
> **Reply:** MoBRAC [28] is an offline reinforcement learning algorithm whose objective is to compute the best response policy for a single agent. It does not aim to compute equilibrium strategies. Because MoBRAC and BOMB target fundamentally different problem formulations, a direct empirical comparison would be meaningless. Our paper already discusses the conceptual distinction between offline RL and offline equilibrium finding, and we include comparisons with two classical offline RL baselines to illustrate this difference. These comparisons further confirm that offline RL methods are not suitable for equilibrium computation in imperfect information extensive form games, which reinforces why MoBRAC is outside the scope of our evaluation.
>
> ----
> Thank you again for the insightful comments. We hope our responses can address your concerns.

---

> > ### Comment · Reviewer_XxFT · 2025-11-27
> >
> > Thank you for the detailed reply. I have a few questions for the authors to clarify to help me better understand the paper.
> >
> > 1. The fundamental contribution of this paper seems to be learning policies directly from a fixed offline dataset, which can be generated by any method as long as every state in the game has appeared. There haven't been previous methods for learning equilibrium policies from offline datasets, right?
> >
> > 2. Figure 2 is somewhat misleading, leading me to believe that this offline dataset is a general dataset containing various games. However, the method in this paper seems to learn from the offline data of each game individually, right?
> >
> > Regarding weaknesses I raised, I need to clarify.
> >
> > Weaknesses 1 & 2: I acknowledge that $\alpha$ doesn't affect the size of the dataset, but the fact that the dataset must cover all states in the game is still a constraint, especially for large-scale games. This makes the motivation somewhat contradictory, as it seems we can only perform this training on small-scale games, which can be solved directly. I hope the authors can clarify the motivation for this work.
> >
> > Weakness 3: I misunderstood before. I think this is actually a major contribution of this paper because it seems to be able to learn policies from datasets generated by any method.
> >
> > Weakness 4: I previously misunderstood that the dataset included various games, but the dataset should actually be each game individually. In this case, this weakness can be ignored.
> >
> > Weakness 5: I think the fundamental difference in this work is learning from offline data.
> >
> > Weakness 6 & 7: I think one contribution of this paper is that using only BC or MB policies cannot guarantee convergence, but BOMB can, right? But in this case, why does the counterexample I gave seem unable to convergence?
> >
> > Question 1: When there are multiple Nash equilibria, how is this gap defined?
> >
> > Overall, I consider the research question addressed in this work to be quite novel, though the current methodology appears to lack practical utility. Nevertheless, I believe the approach to training equilibrium from offline data demonstrates a degree of innovation, hence I raise my score to 4.

---

> > > ### Author Response · Authors · 2025-11-28
> > >
> > > Thanks for the reviewer’s response.
> > >
> > > ----
> > >
> > > ***1: The fundamental contribution of this paper seems to be learning policies directly from a fixed offline dataset, which can be generated by any method as long as every state in the game has appeared. There haven't been previous methods for learning equilibrium policies from offline datasets, right?***
> > >
> > > **Reply:** In the offline equilibrium finding setting, analogous to the offline RL literature, the offline dataset is assumed to be generated by an unknown behavior strategy $\sigma$ as stated in Definition 2. This means that the dataset can be generated by any data collection process and does not require full coverage of all states in the game.
> > >
> > > Regarding prior work, Cui and Du investigated offline equilibrium computation in Markov games, but their study focused on theoretical analysis and did not provide a practical algorithmic solution. In contrast, our work targets imperfect information extensive form games, formally defines the offline equilibrium finding paradigm, constructs datasets for offline evaluation, and proposes a practical algorithm for computing approximate equilibrium strategies from offline data.
> > >
> > > ----
> > >
> > > ***2: Figure 2 is somewhat misleading, leading me to believe that this offline dataset is a general dataset containing various games. However, the method in this paper seems to learn from the offline data of each game individually, right?***
> > >
> > > **Reply:** Yes, our method learns from the offline dataset of each game individually rather than from a single dataset containing multiple games. Figure 2 is intended to illustrate that we study several games and, for each game, construct different types of offline datasets for evaluation.
> > >
> > > ----
> > >
> > > ***Weaknesses 1 & 2: I acknowledge that $\alpha$ doesn't affect the size of the dataset, but the fact that the dataset must cover all states in the game is still a constraint, especially for large-scale games. This makes the motivation somewhat contradictory, as it seems we can only perform this training on small-scale games, which can be solved directly. I hope the authors can clarify the motivation for this work.***
> > >
> > > **Reply:** We would like to clarify that requiring the dataset to cover all states in the game is only an assumption used to establish the theoretical convergence guarantee of our method. This assumption is not required for the practical applicability of the algorithm. In practice, the offline dataset does not need to satisfy full state coverage, and our experiments also operate under partial coverage. The theoretical assumption serves to define a sufficient condition for convergence rather than a condition that must hold in real use cases.
> > >
> > > ----
> > >
> > > ***Weaknesses 6 & 7: I think one contribution of this paper is that using only BC or MB policies cannot guarantee convergence, but BOMB can, right? But in this case, why does the counterexample I gave seem unable to converge?***
> > >
> > > **Reply:**
> > > Our analysis shows that the MB method converges under the uniform coverage assumption, while the BC method converges under the $\epsilon$-equilibrium coverage assumption. Each method works well under different cases, which is precisely why we propose BOMB, whose combination mechanism allows the framework to benefit from both methods.
> > >
> > > Regarding the counterexample, its conclusion depends on the dataset coverage conditions, which were not specified in the example. If the dataset does not cover all game states, then indeed no convergence guarantee applies, as stated in our lower bound analysis (Theorem 4.6). If, instead, the dataset does cover all game states but its sampling distribution shifts away from the optimal policy, then it is important to note that our MB method does not learn a policy from the offline data. It learns the transition function as the environment model, and the equilibrium finding step is carried out by running an online equilibrium algorithm on the trained environment model. Therefore, the online equilibrium algorithm explores the equilibrium strategy through interaction with the trained environment, rather than relying on the offline dataset’s action distribution.
> > >
> > > ----
> > >
> > > ***Question 1: When there are multiple Nash equilibria, how is this gap defined?***
> > >
> > > **Reply:** The gap computation, such as NashConv, does not require specifying any particular Nash equilibrium and depends only on the strategy profile $\sigma$ being evaluated. Therefore, even when multiple Nash equilibria exist, the Nash gap is well defined and does not rely on selecting one equilibrium over another. In addition, the gap in Definition 2 can be instantiated with any standard gap metric, and it is used only as an evaluation measure rather than as part of our method.
> > >
> > > ----
> > > Thank you again for the insightful comments. We hope our responses can address your concerns.

---

### Official Review · Reviewer_wSAG · 2025-10-31

**Soundness:** 3
**Presentation:** 3
**Contribution:** 3
**Rating:** 6
**Confidence:** 3

**Summary:**

The paper introduces "offline equilibrium finding", which is the problem of finding a Nash equilibrium of a (in general, imperfect-information, sequential) game. Various algorithms are studied, including a model-based approach (in which a model is learned from the dataset, and then fed to known equilibrium computation algorithms); a behavioral cloning approach (in which one attempts to train a strategy that directly emulates the strateg(ies) used to generate the data); and a combination of these, which the authors call the Behavior clOning and Model-Based (BOMB) method.

**Strengths:**

The problem of learning a game from an offline dataset seems interesting and important to me. The paper is clear, and the algorithms presented are simple and seemingly effective in experiments, at least in the two-player zero-sum setting.

**Weaknesses:**

* Appendix D raises more questions than it answers for me. If I understand correctly, Cui and Du (2022) showed that, under the unilateral coverage assumption, a Nash equilibrium can be computed in a Markov game, essentially by making pessimistic assumptions about the values in the remainder of the game. But the counterexample in Theorem D.5 *is* a Markov game, if you view the decisions of the two players as simultaneous, which is allowable in this case. Another way of saying this is the following: in the counterexample in Theorem D.5, $(a_1, b_1)$ is always *an* equilibrium, no matter the value of the '?' node. Thus BC works (and I think this implies that BC always works under unilateral coverage assumption?) But it might not be the *unique* equilibrium, so MB does not necessarily work.

  As far as I can tell, this is a (perhaps unnecessary) weakness of the techiques presented in this paper: indeed, Zhang and Sandholm [1] have a similar "pessimism-optimism"-style algorithm for finding equilibria of incomplete extensive-form game models, and in their paper, it certainly suffices for the model ("pseudogame", in their paper) to have what the present paper calls unilateral coverage (and infinite data). I think if you make a similar "pessimistic" assumption, MB should also work under unilateral coverage. Thoughts?

  On a similar note, Zhang and Sandholm [1] have similar ideas to the Cui and Du (2022) paper, for extensive-form games instead of Markov games. Perhaps that is the more relevant citation, since this paper concerns extensive-form games; indeed, I'd like to see some discussion about the results in this paper as compared to [1].

* Experiments, RQ3-4: why not use MB-CFR for CCE computation? CFR is guaranteed to find CCEs, but not Nash, so this feels backwards to me.
* Related question: What is "NashConv" for MB-CFR multi-player games? Is it the Nash gap of the "marginalized" strategy created by taking the marginals of the correlated strategy profile created by CFR? If so, this should be stated explicitly.
* L446: worth mentioning here that computing Nash equilibria, even in normal-form games, is PPAD-hard beyond the two-player zero-sum case (see e.g., [2]), so there are some limitations regarding what one can hope to do efficiently here.

Nitpicks and minor issues:
- "X is guaranteed to be Y if and only if Z" is the same statement as "X is Y if Z". If you mean that "X is Y if Z, and moreover if any of the conditions in Z is broken then X is not necessarily Y" (which you do seem to mean), you should explicitly state the second part as well.
- "States" and "histories", as defined and used by this paper, are basically the same thing, since your definition of "state" basically uniquely identifies a history. I'd pick one of these two words/notations (probably "histories", to align with the EFG literature) and stick to it, for consistency. If for some reason it is relevant that the history is actually represented as a tuple of those items rather than just some abstract representation, you can say something like "We identify histories with tuples of the form ..."
- $\epsilon$-equilibrium coverage is a stronger condition than uniform coverage, right? Because uniform coverage is just the condition $C(s, a) > 0$ for all $(s, a)$?

[1] BH Zhang, T Sandholm (NeurIPS 2020), "Small Nash equilibrium certificates in very large games"

[2] X Chen, X Deng, SH Teng (JACM 2009), "Settling the complexity of computing two-player Nash equilibria"

**Questions:**

1. Perhaps a strange question, but might be good food for thought: are you assuming that the dataset $\mathcal D$ comes from some (possibly correlated) strategy profile? i.e., it is possible that, for two distinct states $s, s'$ in the same information set of a player $i$, the distribution $\mathcal D$ assigns different distributions over the actions at $s$ and $s'$, hence "breaking" the information set. Do you allow this? If so, how do you deal with this, especially in behavioral cloning? If not, have you thought about what would happen if you do allow it?
2. Any responses to anything I've said above?

---

> ### Author Response · Authors · 2025-11-25
> **Response to Reviewer wSAG (1/2)**
>
> We appreciate the reviewer’s comments. Here we provide our response to the weaknesses and your questions.
>
> ----
>
> ***Q: Appendix D raises more questions than it answers for me…***
>
> **Reply:** Thanks for the reviewer insight question.
>
> - In Theorem D.5, indeed, the counter-example can be viewed as a Markov game. However, BC cannot work under a unilateral coverage assumption since BC mimics the underlying behavior strategy of the offline dataset, which means the performance of BC relies on the proportion of some data points. If the dataset satisfies the unilateral coverage assumption, while at every information set the actions appear using the same frequency, for example, the number of datapoints ($I_1, a_1, I_2$) and ($I_1, a_2, I_2$) in the example in Theorem D.5 are the same, then BC will learn a random strategy at information $I_1$. Therefore, BC cannot work under a unilateral coverage assumption.
> - Actually, we have indeed considered adding which assumption that the unilateral dataset coverage assumption would be sufficient for computing the NE strategy in our OEF setting. An assumption is that the NE strategy profile of the game represented by the offline dataset is unique, i.e., the equilibrium uniqueness assumption, which is a very strong assumption. We can add this theorem and proof in the revision if needed.
> - Zhang and Sandholm [1] study how to construct small certificates of approximate Nash equilibrium in very large extensive form games, and they address the equilibrium verification problem instead of the equilibrium finding problem. Thus, while their work is theoretically relevant within extensive form games, it tackles a fundamentally different question.
>
> ----
>
> ***Q: Experiments, RQ3-4: why not use MB-CFR for CCE computation? CFR is guaranteed to find CCEs, but not Nash, so this feels backwards to me.***
>
> **Reply:**
> - In principle, any online equilibrium finding algorithm that can compute a CCE can also be incorporated into the model-based framework. This includes CFR, which is known to converge to a CCE in the standard online setting. Therefore, MB-CFR could indeed be used for CCE computation. However, in our experiments, MB-JPSRO is significantly more efficient in terms of runtime. Since MB-JPSRO also supports CCE computation through its meta solver, it serves as a practical and computationally feasible choice for our evaluation. For these reasons, we adopt MB-JPSRO for CCE computation in our experiments, although MB-CFR remains a valid alternative.
>
> ----
>
> ***Q: Related question: What is "NashConv" for MB-CFR multi-player games? Is it the Nash gap of the "marginalized" strategy created by taking the marginals of the correlated strategy profile created by CFR? If so, this should be stated explicitly.***
>
> **Reply:** The computation of "NashConv" for multi-player games is similar to the computation of two-player games. Formally, $NASHConv(\sigma) = \sum_i (u_i(\sigma^*_{i}, \sigma_{-i})-u_i(\sigma))$. We will state clearly in the revision.
>
> ----
>
> ***Q: L446: worth mentioning here that computing Nash equilibria, even in normal-form games, is PPAD-hard beyond the two-player zero-sum case (see e.g., [2]), so there are some limitations regarding what one can hope to do efficiently here.***
>
> **Reply:**
> - We thank the reviewer for this helpful suggestion. We agree that noting the PPAD hardness of computing Nash equilibria in general normal form games, beyond the two-player zero-sum case, provides useful context for understanding the inherent difficulty of equilibrium computation. We will incorporate this clarification in the revision around Line 446 to make the discussion more complete.
>
> ----
>
> ***Q: "X is guaranteed to be Y if and only if Z" is the same statement as "X is Y if Z"....***
>
> **Reply:** We thank the reviewer for the helpful suggestion and agree with the observation. We will revise the wording in the final version to avoid any misleading appearance of an “if and only if” statement.
>
> ----
>
> ***Q: "States" and "histories", as defined and used by this paper, are basically the same thing…***
>
> **Reply:** Actually, States is slightly different from histories defined in EFG. In our paper, “States” is used in the offline dataset, which represents the game states from the game-level perspective. In Line 159, “States” includes more game information than “Histories”, which only records the past actions.

---

> ### Author Response · Authors · 2025-11-25
> **Response to Reviewer wSAG (2/2)**
>
> ***Q: $\epsilon$-equilibrium coverage is a stronger condition than uniform coverage, right? Because uniform coverage is just the condition $C(s, a) >0$ for all $(s, a)$?***
>
> **Reply:** Actually, uniform coverage and $\epsilon$-equilibrium coverage are not directly comparable in terms of strength, because they constrain different aspects of the data. Uniform coverage requires that $C(s, a) > 0$ for all state-action pairs in the game. This is a very strong assumption, since it demands nonzero coverage over the entire game tree. In contrast, $\epsilon$-equilibrium coverage does not require $C(s, a) > 0$ for every state action pair. Instead, it requires that the offline data provide sufficient visitation and proportion along the trajectories induced by an $\epsilon$-Nash equilibrium. Therefore, the two assumptions enforce different forms of coverage, and neither strictly dominates the other.
>
> ----
>
> ***Q: Perhaps a strange question, but might be good food for thought:...***
>
> **Reply:** We thank the reviewer for the insightful question.
> - In our offline setting, we do not assume that the dataset $D$ is generated by a single strategy profile. For example, a hybrid dataset is a combination of a random dataset and an expert dataset. However, our offline dataset does not contain history level representations (i.e., $s$ and $s'$). Instead, each observation is encoded as a game state vector that corresponds directly to an information state in the underlying extensive form game, as introduced in Section 3.
> - This representation is consistent with the definition of imperfect information extensive form games, in which a player cannot distinguish between histories in the same information set. In our implementation, all histories belonging to an information set are mapped to the same feature representation, so the dataset itself does not differentiate between $s$ and $s'$ at all. For the hybrid dataset, the “breaking” would appear at the information set level instead of the history level.
> - In summary, our dataset is defined at the information state level, not the history level, and therefore, the issue raised does not arise in our offline dataset.
>
> ----
>
> Thank you again for the insightful comments. We hope our responses can address your concerns.

---

> ### Comment · Reviewer_wSAG · 2025-11-25
>
> **Appendix D**
>
> Agh! Pretend I didn't say BC in my review. That was a mis-speak.
>
> What I intended to express was that there *is* a (model-based) algorithm that works in Markov games, namely, the pessimism-based algorithm of Cui and Du. Thus, while it is the case that *your* model-based algorithm does not work in this setting, it does not mean that *no* model-based algorithm works in this setting. Moreover, Zhang and Sandholm [1] gave a similar pessimism-based algorithm for extensive-form games, and both algorithms should *always* work under the unilateral coverage assumption.
>
> I would like to hear your thoughts on this, now that I have (hopefully) expressed myself without silly errors. My apologies.
>
> **MB-CFR for CCEs**
>
> This is rather shocking to me, given that at least some of the games being tested on are small, tabular games, and CFR usually performs *extremely* well in the tabular setting. Did you test modern versions of CFR, such as PCFR+ [3]?
>
> **Nash gap of CFR**
>
> This did not answer my question. Let me be a bit clearer: CFR generates a *sequence* of iterates $\\{\sigma_i^t\\}\_{i \in [n], t \in [T]}$. There are several plausible ways to extract from this a "gap" quantity. Which of these is what you're calling the "Nash gap"?
>
> **(a)** the CCE gap, i.e., the maximum (time-averaged) regret, i.e., $\max_i \max_{\sigma_i'} \frac{1}{T} \sum_{t=1}^T \left[u_i(\sigma_i', \sigma_{-i}^t) - u_i(\sigma^t)\right]$
>
> **(b)** the Nash gap of the marginal strategy profile, $\max_i \max_{\sigma_i'}u_i(\sigma_i', \bar\sigma_{-i}) - u_i(\bar\sigma)$, where the average $\bar\sigma_i = \frac{1}{T} \sum_{t=1}^T \sigma_i^t$ is taken in sequence form. (In zero-sum games, this differs from the CCE gap by only a constant factor.)
>
> **(c)** the Nash gap of the last iterate, $\max_i \max_{\sigma_i'}u_i(\sigma_i', \sigma_{-i}^T) - u_i(\sigma^T)$
>
> **(d)** Something else? (If so, what?)
>
> Note that, in zero-sum games, (a) and (b) are guaranteed to converge to 0; in non-zero sum games, only (a) is guaranteed to converge to 0.
>
> Depending on your answer, I have follow-up questions:
>
> If you are using (a), you should not call it a Nash gap, because it is a CCE gap, not a Nash gap. If this is the case, what's the distinction between RQ3 and RQ4?
>
> If you are using (b) or (c), you should note that CFR is not guaranteed to converge to Nash equilibria in non-zero sum games, only to CCEs. (This is why I found it very strange that you were using CFR in the Nash section but not in the CCE section!)
>
> I hope this clarifies my question.
>
> **States and histories**
>
> I see. I must say that this is a rather unusual usage of the word "state". In the RL literature, a state typically only encodes information relevant for the future, not the history of past actions or players' information sets. I would use a different word here, perhaps "history representation". I would also not mind simply overloading the term "histories"---I know that those tuples are not technically histories, but they serve the purpose of uniquely identifying a history, so they're basically the same thing in my book.
>
> **$\epsilon$-equilibrium coverage**
>
> But Definition 4.2 demands that $\sigma_D(s_t, a_t) > 0$ for all $s_t, a_t$. Doesn't this, along with the large-dataset assumption, imply uniform coverage?
>
> (Also, apologies, my initial reply contained a few typos, which hopefully have been fixed before they were noticed :) )
>
> [3] G Farina, C Kroer, T Sandholm (AAAI 2021) "Faster game solving via predictive Blackwell approachability: Connecting regret matching and mirror descent"

---

> > ### Author Response · Authors · 2025-11-26
> >
> > Thanks for your prompt response! For your concerns, our responses are listed below:
> >
> > ----
> >
> > ***Appendix D***
> >
> > **Reply:** Thank you for the clarification and for raising this important point.
> > - Markov games assume full state observability and admit Bellman-style value recursion, making a pessimism-based algorithm feasible. However, imperfect-information extensive-form games involve information sets that aggregate multiple indistinguishable histories, which break the Bellman structure. As a result, pessimism-based algorithms developed for Markov games cannot directly extend to the IIEFG setting.
> > - The pseudogame framework of Zhang and Sandholm provides powerful tools for equilibrium verification when upper and lower payoff bounds for each information set are already available. However, their work does not address how such payoff bounds can be learned from finite offline data, nor how to handle partial coverage. Without these bounds, pseudogame pessimism cannot be directly instantiated in the offline learning setting.
> > - We agree that our model-based approach does not succeed under the unilateral coverage assumption, but this does not imply that no model-based approach could succeed. Designing a pessimism-based offline equilibrium finding algorithm for IIEFGs would be a promising and interesting direction for future exploration.
> >
> > ----
> >
> > ***MB-CFR for CCEs***
> >
> > **Reply:** Thank you for your question.
> > - We would like to clarify that the focus of our work is the model-based framework. The main novelty lies in training an environment model from an offline dataset and then applying an online equilibrium finding algorithm to this trained environment model. The framework is intentionally designed to be flexible and modular, and any online algorithm, including CFR and modern variants such as PCFR+, is fully compatible with it.
> > - We fully agree that CFR-based methods are strong online algorithms capable of computing CCEs. In our experiments, however, the goal is not to benchmark different online algorithms, but rather to study how the model-based framework behaves when using a trained environment model in the offline setting. In this context, the online solver functions as a plug-in component rather than the primary object of study.
> > - Moreover,  as illustrated in Figure 19, the performance of the model-based framework largely depends on the trained environment model rather than the specific choice of online equilibrium finding algorithm. MB-JPSRO is therefore used as one concrete instantiation within a general and widely applicable framework, and other online algorithms could be incorporated without changing the methodology.
> >
> > ----
> >
> > ***Nash gap of CFR***
> >
> > **Reply:** Thank you for the detailed follow-up. We apologize for the earlier lack of clarity. In our experiments, the quantity we report as the “Nash gap” for CFR corresponds to (b) in your list. Our goal in RQ3 and RQ4 is not to evaluate the convergence properties of specific online algorithms, but rather to demonstrate that the offline equilibrium finding (OEF) framework is flexible and can be paired with different online algorithms to approximate different equilibrium concepts. Thus, the choice of online algorithms affects only the approximate equilibrium computed within the model-based framework, but does not reflect a claim about the convergence guarantees of CFR in multiplayer games.
> >
> > ----
> >
> > ***States and histories***
> >
> > **Reply:** Thanks for your comments. We agree that our use of the term “state” may not align with the conventional terminology in the RL literature. We appreciate the reviewer’s suggestion and will revise the terminology in the paper to avoid confusion.
> >
> > ----
> >
> > ***$\epsilon$-equilibrium coverage***
> >
> > **Reply:** Thank you for pointing this out. You are correct that $\epsilon$-equilibrium coverage assumption is indeed stronger than uniform coverage. Our earlier response was intended to highlight the different focuses of the two assumptions. We agree that our previous explanation did not clearly acknowledge the relative strength of the $\epsilon$-equilibrium coverage assumption and may have caused confusion.
> >
> > ----
> >
> > Thank you again for the prompt response. We hope our responses can address your concerns.

---

> > > ### Comment · Reviewer_wSAG · 2025-11-26
> > >
> > > **Appendix D**
> > >
> > > > Markov games assume full state observability and admit Bellman-style value recursion, making a pessimism-based algorithm feasible. However, imperfect-information extensive-form games involve information sets that aggregate multiple indistinguishable histories, which break the Bellman structure. As a result, pessimism-based algorithms developed for Markov games cannot directly extend to the IIEFG setting.
> > >
> > > Agreed.
> > >
> > > > The pseudogame framework of Zhang and Sandholm provides powerful tools for equilibrium verification when upper and lower payoff bounds for each information set are already available. However, their work does not address how such payoff bounds can be learned from finite offline data, nor how to handle partial coverage.
> > >
> > > Agreed, and I am not at all claiming that Cui-Du nor Zhang-Sandholm subsume the present paper. I am, however, claiming the following: once you *have* some approximated game from offline data, the pessimism-based technique of Zhang-Sandholm (which seems to me to be the extensive-form analogue of the pessimism-based algorithm of Cui-Du) can be used as a drop-in replacement for your model-based algorithm, and it works under the unilateral coverage assumption. Let me attempt to be a bit more formal:
> > >
> > > Suppose we have a sufficiently large dataset for a game $G$, satisfying the unilateral coverage assumption under some profile $\sigma^\*$. Then the "approximated game" $\tilde G$ consists of all nodes that can be touched if one player deviates from this strategy profile. The game $\tilde G$ therefore has sufficient information to find some Nash equilibrium, namely $\sigma^*$, if only one could write down an algorithm that achieves this.
> > >
> > > Let's write down such an algorithm, at least for zero-sum games. Let $\tilde G^-$ be the game in which all nodes not touched by the dataset are replaced by terminal nodes with value $-L$ for the MAX-player, where $L$ is the largest magnitude utility in the original game. Let $\tilde G^+$ be the same, except with $-L$ replaced by $+L$. Then one can show the following: Under the unilateral coverage assumption, $\tilde G^+$ and $\tilde G^-$ have the same value. Moreover, in this case, if $x^-$ is a Nash strategy for MAX in $\tilde G^-$, and $y^+$ is a Nash strategy for MIN in $\tilde G^+$, then $(x^-, y^+)$ is a Nash strategy in the original game $G$.
> > >
> > > This gives a model-based learning algorithm, at least for zero-sum games, under the unilateral coverage assumption: Build the "bounding" games $\tilde G^-$ and $\tilde G^+$, solve them both (using any equilibrium finding algorithm of your choice), and finally extract and return the profile $(x^-, y^+)$. (In the language of Zhang-Sandholm, $\tilde G$ is a trunk, or a pseudogame, and $(x^-, y^+)$ is a certificate of equilibrium in $G$.)
> > >
> > > > Designing a pessimism-based offline equilibrium finding algorithm for IIEFGs would be a promising and interesting direction for future exploration.
> > >
> > > I think the above argument solves this problem, at least for the zero-sum case.
> > >
> > >
> > > **CFR for Nash**
> > >
> > > I guess the thing that confused me was the following:
> > > * You use CFR to find Nash. CFR is not guaranteed to find Nash.
> > > * You do not use CFR to find CCEs. CFR is guaranteed to find CCEs.
> > >
> > > This seems backwards. What gives?

---

> > > > ### Author Response · Authors · 2025-11-27
> > > >
> > > > Thanks for the reviewer’s prompt and detailed response.
> > > >
> > > > ----
> > > >
> > > > ***Appendix D***
> > > >
> > > > We thank the reviewer for the detailed description, and these comments are constructive for our future work.
> > > >
> > > > ----
> > > >
> > > > ***CFR for Nash***
> > > >
> > > > From a theoretical perspective, we agree that the experimental choices may appear inverted, and we appreciate the opportunity to clarify our intention. In designing the experiments, our primary goal was to demonstrate the flexibility and generality of the model-based framework.
> > > >
> > > > In RQ3, we used CFR to compute NE in two-player games, and we carried this choice over when extending to multiplayer games. In practice, CFR is widely used for multiplayer imperfect information games, even though it does not offer NE convergence guarantees, since existing methods do not provide general convergence guarantees for NE computation in multiplayer settings. Our intention was therefore to adopt a commonly used practical algorithm within the model-based framework.
> > > >
> > > > For the CCE experiments, we intended to highlight that the model-based framework can seamlessly incorporate different online algorithms. Since JPSRO can compute CCEs and is widely used, we selected it to illustrate the flexibility of the model-based algorithm. In hindsight, we agree that this choice may look unusual when contrasted with the Nash experiments, and we appreciate the reviewer for pointing this out. We will revise the paper to make our motivation clearer.
> > > >
> > > > ----
> > > >
> > > > Thank you again for the prompt response. We hope our responses can address your concerns.

---

### Official Review · Reviewer_nYzC · 2025-11-01

**Soundness:** 2
**Presentation:** 2
**Contribution:** 2
**Rating:** 4
**Confidence:** 4

**Summary:**

This paper proposes a hybrid framework, BOMB, for offline equilibrium finding in extensive-form games. The core idea is to mix the behavior cloning (BC) policy from the offline dataset with the policy learned by online model-based (MB) equilibrium-learning algorithms, including CFR and PSRO. The authors provide a dataset with random or expert game trajectories and learn game models for subsequent online learning. Theoretically, they provide necessary and sufficient conditions (about data coverage) for the MB and BC policies to approach equilibrium. Through experiments, they verify that under a learning-based estimation of the policy mixing parameter, BOMB outperforms BC, MB, and two applicable offline RL algorithms.

**Strengths:**

1. It is important to examine the open problem of offline equilibrium finding, especially under the imperfect-information setting.

2. This paper makes significant efforts in creating offline datasets and demonstrating the advantage of using a hybrid policy in the offline setting, providing both theoretical and empirical analysis.

**Weaknesses:**

1. From my perspective, the advantage of BOMB mainly comes from the existence of a learnable mixing parameter. However, the paper does not explain how the learning-based estimation method (Figure 3) works. Estimating the mixing parameter simply based on a similarity vector is intuitively unsound, since it completely ignores the game structure or the distribution of offline data.

2. While Theorem 4.4 and Theorem 4.5 seem correct to me, the proof of Theorem 4.6 is quite insufficient. Line 1308 suggests that Theorem 4.4 guarantees the Nash gap of the MB policy is smaller than that of the BC policy, which is not true even in the authors' own experiment of 5-player Kuhn poker (Figure 8). Even if the statement is correct, arbitrarily mixing the two policies could result in a performance drop rather than improvement without further theoretical analysis.

3. The authors claim that the MB framework’s performance is shown to be independent of the specific algorithm used. This is quite counterintuitive and only verified in the simplest 2-player Kuhn poker for CFR and PSRO (Figure 19).

4. The typos should be checked and corrected in the paper. For example, on Page 4, it should be "Liar's Dice" in Figure 2, "state" in Line 174, and "CFR (Zinkevich et al., 2007)" in Line 204.

**Questions:**

See Weaknesses.

---

> ### Author Response · Authors · 2025-11-25
> **Response to Reviewer nYzC**
>
> We appreciate the reviewer’s comments. Here we provide our response to the weaknesses and your questions.
>
> ----
>
> ***Q: From my perspective, the advantage of BOMB mainly comes from the existence of a learnable mixing parameter…***
>
> **Reply:**
> - We would like to clarify the rationale behind our design choice. In the offline equilibrium finding setting, only the offline dataset is available, and the underlying strategy profile that generated this dataset is not available. As a result, information about the true game structure cannot be used for estimating the combination parameter. In fact, the BC strategy directly reflects the empirical distribution contained in the offline dataset.
> - Under these constraints, we use the CKA similarity vector computed between the BC policy and the MB policy as the input for estimating the combination coefficient. The predictor is trained in a supervised manner to output the value of $\alpha$ given this CKA similarity vector. Although the training of this predictor uses the optimal $\alpha$ as labels, which may require simulator access, once the predictor is trained, it can be applied entirely in a fully offline setting. Empirically, this method is effective and consistently improves performance, as demonstrated in our experiments.
>
> ----
>
> ***Q: While Theorem 4.4 and Theorem 4.5 seem correct to me, the proof of Theorem 4.6 is quite insufficient…***
>
> **Reply:**
> - In the proof of Theorem 4.6, Line 1308 states that if the dataset satisfies the uniform coverage assumption, then the Nash gap of the MB policy is equal to or smaller than that of the BC policy. Theorem 4.6 provides a lower bound on the performance of BOMB, namely that its performance is at least as good as that of BC. In the 5-player Kuhn Poker experiment (Figure 8), we indeed observe that the performance of BOMB is equal to or better than that of BC.
> - When the proportion of hybrid data equals 1.0, the hybrid dataset reduces to the random dataset. However, the random dataset for 5-player Kuhn Poker does not guarantee the uniform coverage assumption, and the training of the environment model also introduces approximation error, which is not considered in the theoretical analysis. These factors explain why MB does not always outperform BC in this specific setting. Notably, when the dataset size increases (2e4), MB outperforms BC under the hybrid dataset with a proportion of 1.0. This aligns with the theoretical requirement that sufficient coverage and accurate environment modelling are needed for the MB policy to achieve better performance.
>
> ----
>
> ***Q: The authors claim that the MB framework’s performance is shown to be independent of the specific algorithm used. This is quite counterintuitive and only verified in the simplest 2-player Kuhn poker for CFR and PSRO (Figure 19).***
>
> **Reply:**
> - This claim is made from the model-based framework. Compared to traditional online equilibrium finding algorithms, our model-based framework only replaces the real environment with the trained environment model. The only difference is the environment. Therefore, the performance of the model-based framework mainly depends on the quality of the trained environment model instead of the online equilibrium finding algorithm used in the model-based framework, since we did not modify the process of the online equilibrium finding algorithm used in the model-based framework.
>
> ----
>
> ***Q: The typos should be checked and corrected in the paper.***
>
> **Reply:** Thanks for pointing that out. We will modify them in the revision.
>
> ----
>
> Thank you again for the insightful comments. We hope our responses can address your concerns.

---

### Official Review · Reviewer_yN9Z · 2025-11-05

**Soundness:** 3
**Presentation:** 3
**Contribution:** 2
**Rating:** 6
**Confidence:** 2

**Summary:**

This paper argues that offline equilibrium finding is under-explored, and proposes new datasets, a new method (BOMB), and experiments validating the new method.

**Strengths:**

The research direction is promising and several experiments are performed.

**Weaknesses:**

I wish there was more on the related works.

It is mentioned that there is existing work that does offline learning in Markov games. I would have appreciated more insight into this previous research and why or why not it doesn't work for EFGs.

I would have appreciated more insight into previous research on real-world games like football, such as papers by Karl Tuyls et al. like "Game Plan: What AI can do for Football, and What Football can do for AI": how does this line of research relate or not relate to offline equilibrium finding?

And there do seem to be some papers that come up when one searches for "offline multi-agent reinforcement learning". How do these apply or not apply to offline equilibrium finding?

**Questions:**

see "weaknesses"

---

> ### Author Response · Authors · 2025-11-25
> **Response to Reviewer yN9Z**
>
> We appreciate the reviewer’s comments. Here we provide our response to the weaknesses and your questions.
>
> ----
>
> ***Q: I wish there was more on the related works. It is mentioned that there is existing work that does offline learning in Markov games. I would have appreciated more insight into this previous research and why or why not it doesn't work for EFGs.***
>
> **Reply:**
> - Existing works on offline learning in Markov games indeed provide valuable theoretical insights, and we build on some of these ideas in our own analysis. However, most of these works focus on the fully observable Markov game setting and do not provide practical algorithms for imperfect information extensive form games.
>
> - The structural differences between Markov games and IIEFGs are substantial: Markov games assume fully observable states and do not involve information sets. The information set component is essential in IIEFGs and makes the offline setting significantly more challenging. For this reason, the theoretical results for offline Markov games cannot be directly transferred to IIEFGs (as shown in our theoretical analysis in Appendix D), and practical algorithms are required.
>
> ----
>
> ***Q: I would have appreciated more insight into previous research on real-world games like football, such as papers by Karl Tuyls et al., like "Game Plan: What AI can do for Football, and What Football can do for AI": how does this line of research relate or not relate to offline equilibrium finding?***
>
> **Reply:**
> - Thank you for highlighting this line of work. Papers such as “Game Plan: What AI Can Do for Football, and What Football Can Do for AI” indeed analyze multi-agent interactions from offline observational data and study whether players’ behaviors are consistent with equilibrium concepts. In that sense, they are slightly related to our offline setting.
> - However, this paper is more like EGTA. They aim to infer payoff structures or evaluate strategic behavior in real-world domains using statistical modeling, computer vision, or domain-specific analytics. Critically, they do not address the algorithmic problem of computing equilibria from offline data. Their focus is on behavioral analysis of human or agent trajectories, rather than designing general-purpose offline equilibrium-finding methods.
>
> ----
>
> ***Q: And there do seem to be some papers that come up when one searches for "offline multi-agent reinforcement learning". How do these apply or not apply to offline equilibrium finding?***
>
> **Reply:**
> - Indeed, there are many works on offline multi-agent reinforcement learning, but these methods mainly focus on cooperative settings, where the goal is to maximize the joint return of all agents. In contrast, offline equilibrium finding in IIEFGs addresses fundamentally competitive settings such as zero-sum or general-sum games, where the goal is to compute an equilibrium strategy. Therefore, offline MARL algorithms cannot be directly applied to our offline equilibrium-finding setting.
>
> ----
>
> Thank you again for the insightful comments. We hope our responses can address your concerns.

---

### Author Response · Authors · 2025-11-28

Thanks to all the reviewers' comments. We have made the following revisions in response to your comments:

- Clarified the computation of NASHCONV in multi-player games in Section 2.
- Clarified the use of the “state” representation in Section 3.
- Revised the theoretical statements in Section 4 to remove the “if and only if” implication.
- Added an explanation of the limitations of Nash computation in multi-player settings in RQ3, Section 5.
- Fixed the typos in Figure 2 and on Page 4.

We hope that our responses and modifications can address your concerns!

---

### Meta-Review · Area_Chair_uiWJ · 2026-01-05

**Summary:**

Strength: The paper considers the offline EF problem, which is underexplored. The experimental results show effectiveness for the proposed method.

Weakness: The analysis assumes strong conditions. The experimental setup could be strengthened. Also, more recent related works should be discussed and compared with.

**Reviewer Concerns:**

Weakness: The analysis assumes strong conditions. The experimental setup could be strengthened. Also, more recent related works should be discussed and compared with.

**Reviewer Scores:**

Based on the comments and discussions, the reviewers will likely keep their scores.

---

### Decision · Program_Chairs · 2026-01-26

Reject